# TANDEM: Bi-Level Data Mixture Optimization with Twin Networks

**Jiaxing Wang**[*1], **Deping Xiang**[*1], **Jin Xu**[2], **Mingyang Yi**[†3], **Guoqiang Gong**[1],
**Zicheng Zhang**[1], **Haoran Li**[4], **Pengzhang Liu**[†1], **Zhen Chen**[1], **Ke Zhang**[1],
**Ju Fan**[3], **Qixiang Jiang**[1]

[1]JD.com    [2] University of Oxford    [3]Renmin University of China
[4]University of Chinese Academy of Sciences
{wangjiaxing41,xiangdeping1,gongguoqiang1,zhangzicheng6}@jd.com
{liupengzhang,chenzhen48,zhangke323,jiangqixia}@jd.com
{yimingyang,fanj}@ruc.edu.cn
jin.xu@stats.ox.ac.uk
lihaoran21@mails.ucas.ac.cn

## Abstract

The capabilities of large language models (LLMs) significantly depend on training data drawn from various domains. Optimizing domain-specific mixture ratios can be modeled as a bi-level optimization problem, which we simplify into a single-level penalized form and solve with twin networks: a proxy model trained on primary data and a dynamically updated reference model trained with additional data. Our proposed method, Twin Networks for bi-level DatA mixturE optiMization (TANDEM), measures the data efficacy through the difference between the twin models and up-weights domains that benefit more from the additional data. TANDEM provides theoretical guarantees and wider applicability, compared to prior approaches. Furthermore, our bi-level perspective suggests new settings to study domain reweighting such as data-restricted scenarios and supervised fine-tuning, where optimized mixture ratios significantly improve the performance. Extensive experiments validate TANDEM's effectiveness in all scenarios.

## 1  Introduction

The success of large language models (LLMs) largely relies on extensive training data collected from diverse domains, including chat logs [18], academic writings [32], mathematical problems [45], and code repositories [21]. The emergent capabilities observed in LLMs are substantially influenced by the specific composition of cross-domain corpora [13, 38]. Therefore, it is important to carefully balance the proportions of domain-specific data in training sets to ensure models develop intended and balanced capabilities for target domains.

Optimizing the mixture ratios of data domains can be formulated as a bi-level optimization problem, in which the inner loop optimizes model parameters for a fixed ratio on training data, while the outer loop searches for the best mixture ratio on validation data. Due to the difficulty of exactly solving this bi-level problem, we transform it into a single-level optimization problem. Specifically, the inner-level optimization is viewed as a Lagrangian penalty within the outer-level objective. This perspective naturally motivates the introduction of twin networks: a proxy model trained exclusively on the primary training data, and a reference model exposed to additional validation data. Interestingly, this twin-network formulation relates closely to prior methods such as DoReMi [37] and DoGE

---

[*]Equal Contribution.
[†]Corresponding to Mingyang Yi and Pengzhang Liu

39th Conference on Neural Information Processing Systems (NeurIPS 2025).

[7], offering insights into addressing their limitations. Based on this formulation, we propose Twin Networks for bi-level DatA mixturE optiMization (TANDEM), an algorithm that measures the domain data efficacy through the twin models' disparities and ultimately approximates the original bi-level optimization problem. Unlike DoReMi, which employs a static reference model, TANDEM dynamically updates both models. Additionally, TANDEM enhances stability and provides theoretical convergence guarantees compared to DoGE, as it aggregates multiple updating steps and avoids relying directly on gradient estimation, thereby mitigating issues related to high variance.

The bi-level formulation also emphasizes that future research on data domain weighting could focus more on scenarios with limited domain-specific data and supervised fine-tuning (SFT), rather than exclusively on traditional pretraining settings with abundant data. From a bi-level optimization viewpoint, assigning equal mixture ratios becomes a valid solution for single-epoch training when domain data is abundant, aligning with recent empirical findings highlighting uniform mixing strategies as competitive baselines [2]. Despite the prevalence of big data, limited data scenarios are quite common, particularly within specific domains, since large datasets frequently consist of many heterogeneous smaller datasets [33]. Furthermore, SFT often requires domain data to be visited multiple times, which creates generalization gap that leads to non-trivial solution for the bi-level problem. It is precisely in these cases that optimizing data mixture ratios can yield significant improvements.

While previous methods like DoReMi and those built on data mixing laws [41, 23, 16] are not directly applicable to these newer scenarios, TANDEM can be effectively extended. Our experimental results demonstrate TANDEM's effectiveness in such settings.

Our contributions can be summarized as follows:

- We introduce TANDEM, an effective and efficient algorithm for data mixture optimization that utilizes twined proxy and reference networks to approximate the bi-level objective. TANDEM enjoys theoretical convergence guarantees.
- We highlight that data mixture optimization is particularly beneficial in scenarios with limited data availability rather than traditional pretraining setups with abundant domain data.
- We empirically show TANDEM's effectiveness across standard and data-limited scenarios, showing its superiority over a set of competitive data mixture optimization methods.

## 2   Methodology

We formulate the problem of finding the optimal data mixture ratio as bi-level optimization. To solve the problem, we propose our penalty-based algorithm TANDEM, as presented in Figure 1(b) and Algorithm 1 in Appendix A. TANDEM draws insights from many previous works, while improving upon them. Besides, by inspecting the bi-level optimization formulation, we suggest broadening the scope of research on data mixture optimization under limited-domain data and supervised fine-tuning (SFT), rather than focusing solely on conventional data-rich pretraining settings. The notations used in this paper are summarized in Appendix H.

### 2.1   Problem Formulation

Consider training an LLM on a data composition from $M$ domains, $\{\mathcal{D}_1, \mathcal{D}_2, \ldots, \mathcal{D}_M\}$ (e.g. Wikipedia, CommonCrawl). Data mixture optimization (DMO) refers to the problem of finding the optimal proportions of data for each domain $\boldsymbol{\alpha} = [\alpha_1, \alpha_2, \ldots, \alpha_M]$ over probability simplex $\mathcal{A} := \{\boldsymbol{\alpha} \in \mathbb{R}^M \mid \sum_{m=1}^{M} \boldsymbol{\alpha}_m = 1, \alpha_m \geq 0\}$. For any data mixture ratio $\boldsymbol{\alpha}$, and its corresponding model parameter $\boldsymbol{w}(\boldsymbol{\alpha})$ obtained by training loss, our goal is to minimize the validation loss $\mathcal{L}_{\text{val}}(\boldsymbol{w}(\boldsymbol{\alpha}))$ over $\boldsymbol{\alpha}$. The optimization problem is formulated as follows:

$$\min_{\boldsymbol{\alpha} \in \mathcal{A}} \mathcal{L}_{\text{val}}(\boldsymbol{w}(\boldsymbol{\alpha})) := \sum_{m=1}^{M} \mathcal{L}_{\text{val}}^m(\boldsymbol{w}(\boldsymbol{\alpha})) \quad \text{s.t.} \ \boldsymbol{w}(\boldsymbol{\alpha}) \in \arg\min_{\boldsymbol{w}} \mathcal{L}_{\text{train}}(\boldsymbol{\alpha}, \boldsymbol{w}) := \sum_{m=1}^{M} \boldsymbol{\alpha}_m \mathcal{L}_{\text{train}}^m(\boldsymbol{w}). \quad (1)$$

By splitting the data into training and validation sets, we can construct the validation and training loss $\mathcal{L}_{\text{val}}^m(\boldsymbol{w})$ and $\mathcal{L}_{\text{train}}^m(\boldsymbol{w})$ on domain $\mathcal{D}_m$. Intuitively, in the bi-level problem, the outer level problem seeks the optimal domain weights for validation loss with the model weights obtained by the inner level reweighted training loss. Similar to [37, 7], given the learned final mixture ratio $\boldsymbol{\alpha}^*$,

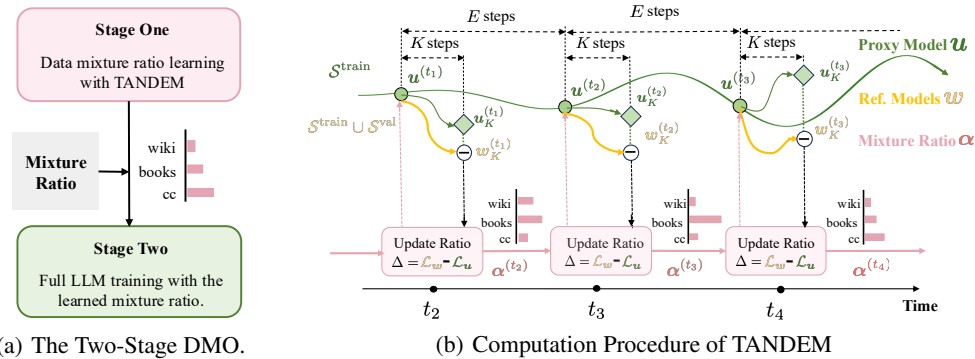

(a) The Two-Stage DMO.  (b) Computation Procedure of TANDEM

Figure 1: (a) The two-stage data mixture optimization. Optimal mixtures are first learned and then utilized to train the final model. (b) The computation procedure of TANDEM, twined proxy model (green) and reference model (orange) are used to determine the update of the mixture ratio (pink).

we construct the final training set by sampling $\mathcal{D}_{\boldsymbol{\alpha}^*} \triangleq \sum_{m=1}^M \boldsymbol{\alpha}^* \cdot \mathrm{UNIF}\left(\mathcal{D}_m\right)$ upon which the final model is trained (Figure 1(a)).

## 2.2 Twin Networks for Bi-level Data Mixture Optimization

Solving the bi-level optimization problem (1) is challenging due to its nested structure. By viewing the inner-level problem as a constraint and subsequently incorporating it into the outer-level problem as a Lagrangian penalty, (1) can be reformulated as a single-level problem [30, 19]:

$$\min_{\boldsymbol{\alpha} \in \mathcal{A}, \boldsymbol{w}} \mathcal{H}_\gamma(\boldsymbol{\alpha}, \boldsymbol{w}) := \mathcal{L}_{\mathrm{val}}(\boldsymbol{w}) + \gamma \left( \mathcal{L}_{\mathrm{train}}(\boldsymbol{\alpha}, \boldsymbol{w}) - \min_{\boldsymbol{u}} \mathcal{L}_{\mathrm{train}}(\boldsymbol{\alpha}, \boldsymbol{u}) \right). \quad (2)$$

Here, the auxiliary variable $\boldsymbol{u}$ is introduced as a proxy of $\boldsymbol{w}(\boldsymbol{\alpha}) \in S^*(\boldsymbol{\alpha}) := \arg\min_{\boldsymbol{w}} \mathcal{L}_{\mathrm{train}}(\boldsymbol{\alpha}, \boldsymbol{w})$. The constrain in (1) is transferred into the penalization $\mathcal{L}_{\mathrm{train}}(\boldsymbol{\alpha}, \boldsymbol{w}) - \min_{\boldsymbol{u}} \mathcal{L}_{\mathrm{train}}(\boldsymbol{\alpha}, \boldsymbol{u})$. Clearly, by properly invoking $\gamma \to \infty$, the solution of (2) will approximate the original (1). This claim is justified by Proposition 3 in [30]. We refer readers to the Appendix A for more information. Next, we proceed to illustrate our algorithm of optimizing the penalized Lagrange problem (2).

**Algorithm Procedure**  Besides the mixture ratio $\boldsymbol{\alpha}$, optimizing (2) deals with two models: a proxy model $\boldsymbol{u}$ and a reference model $\boldsymbol{w}$. As shown in Figure 1(b), the optimization processes of $\boldsymbol{\alpha}$, $\boldsymbol{w}$, and $\boldsymbol{u}$ are indexed by $t$, $k$ and $k$.

***Update on $\boldsymbol{u}$***: Firstly, given a data mixture ratio $\boldsymbol{\alpha}^{(t)}$, we find the $\boldsymbol{u} \in \arg\min_{\boldsymbol{u}} \mathcal{L}_{\mathrm{train}}(\boldsymbol{\alpha}^{(t)}, \boldsymbol{u})$ in the penalization term under certain mixture ratio $\boldsymbol{\alpha}$. Our $\boldsymbol{u}$ is updated for $K$ steps to approximate the optimal one:

$$\boldsymbol{u}_{k+1}^{(t)} = \boldsymbol{u}_k^{(t)} - \eta_u \nabla_{\boldsymbol{u}} \mathcal{L}_{\mathrm{train}}\left(\boldsymbol{\alpha}^{(t)}, \boldsymbol{u}_k^{(t)}\right) \quad (3)$$

(green line with arrow in Figure 1(b)). The proxy model is trained on the whole training set and maintained through the DMO process.

***Update on $\boldsymbol{w}$***: $\boldsymbol{w}$ serves as a reference model updated on both the training and the validation sets. Similar to the proxy model $\boldsymbol{u}$, $\boldsymbol{w}$ is updated for $K$ steps before one data mixture ratio $\boldsymbol{\alpha}$ update, which is used to optimize the inner problem (2). Note that the term related to $\boldsymbol{w}$ is $\mathcal{L}_{\mathrm{val}}(\boldsymbol{w}) + \gamma \mathcal{L}_{\mathrm{train}}(\boldsymbol{\alpha}, \boldsymbol{w})$, the update rule then becomes:

$$\boldsymbol{w}_{k+1}^{(t)} = \boldsymbol{w}_k^{(t)} - \eta_{\boldsymbol{w}} \left( \nabla_{\boldsymbol{w}} \mathcal{L}_{\mathrm{val}}\left(\boldsymbol{\alpha}^{(t)}, \boldsymbol{w}_k^{(t)}\right) + \gamma \nabla_{\boldsymbol{w}} \mathcal{L}_{\mathrm{train}}\left(\boldsymbol{\alpha}^{(t)}, \boldsymbol{w}_k^{(t)}\right) \right), \quad (4)$$

(orange line in Figure 1(b)). Intuitively, training $\boldsymbol{w}$ for multiple steps provides more subtle guidance for mixture ratio update. This will be elaborated in the $\boldsymbol{\alpha}$ update part.

Unlike the proxy model $\boldsymbol{u}$, which is maintained throughout training (green line in Figure 1(b)), we do not maintain an independent reference model $\boldsymbol{w}$, but rather synchronize the starting point of $\boldsymbol{w}$ and $\boldsymbol{u}$ by setting $\boldsymbol{w}_0^{(t)} = \boldsymbol{u}_0^{(t)}$ as a initialization of the $K$ updates. By doing so, we control the disparity

between $\boldsymbol{w}$ and $\boldsymbol{u}$ during the optimization. Intuitively, $\boldsymbol{w}$ and $\boldsymbol{u}$ should not diverge from each other as $\boldsymbol{u}$ acts as a proxy of $\boldsymbol{w}(\boldsymbol{\alpha}) \in S^*(\boldsymbol{\alpha})$ and $\boldsymbol{w}$ approximates $\boldsymbol{w}_\gamma^*(\boldsymbol{\alpha}) \in S_\gamma^*(\boldsymbol{\alpha}) := \arg\min_{\boldsymbol{w}} \mathcal{H}_\gamma(\boldsymbol{\alpha}, \boldsymbol{w})$. Clearly, the ideal $\boldsymbol{w}_\gamma^*(\boldsymbol{\alpha})$ under penalized problem will approximate one $\boldsymbol{w}(\boldsymbol{\alpha})$ when $\gamma \to \infty$, since the penalized Lagrange problem (2) approximates the original problem (1).

***Update on $\boldsymbol{\alpha}$***: The mixture ratio update in (5) seeks a solution of (2). That says, a data mixture ratio yields a trained model with good performance on data from the validation set under all domains. The term in (2) related to $\boldsymbol{\alpha}$ is $\gamma \left(\mathcal{L}_{\text{train}}(\boldsymbol{\alpha}, \boldsymbol{w}) - \min_{\boldsymbol{u}} \mathcal{L}_{\text{train}}(\boldsymbol{\alpha}, \boldsymbol{u})\right)$. Recall that $\mathcal{L}_{\text{train}}(\boldsymbol{\alpha}, \cdot)$, by definition, is the $\boldsymbol{\alpha}$ weighted domain-wise loss. Applying the projected gradient descent gives the following update rule:

$$\boldsymbol{\alpha}^{(t+1)} = \Pi_{\mathcal{A}} \left( \boldsymbol{\alpha}^{(t)} - \eta_{\boldsymbol{\alpha}} \gamma \left( \underbrace{\mathcal{L}_{\text{train}}^{1:M}\left(\boldsymbol{\alpha}^{(t)}, \boldsymbol{w}_K^{(t)}\right)}_{\text{reference model}} - \underbrace{\mathcal{L}_{\text{train}}^{1:M}\left(\boldsymbol{\alpha}^{(t)}, \boldsymbol{u}_K^{(t)}\right)}_{\text{proxy model}} \right) \right), \qquad (5)$$

(Pink line in Figure 1(b).) where $\Pi_{\mathcal{A}}(\cdot)$ projects the updated mixture ratio to the probabilistic simplex. The post-updated $\boldsymbol{u}_K^{(t)}$ and $\boldsymbol{w}_K^{(t)}$ are applied to capture the domain-wise loss difference $\mathcal{L}_{\text{train}}^m(\boldsymbol{w}_K^{(t)}) - \mathcal{L}_{\text{train}}^m(\boldsymbol{u}_K^{(t)})$. Since the $\boldsymbol{u}_K^{(t)}$ and $\boldsymbol{w}_K^{(t)}$ are respectively trained on training set and training set plus validation set, the aforementioned gap captures the gain of incorporating the additional validation data. Larger loss differences indicate that the model gets significant improvement by consuming more data, thus the corresponding domains are up-weighted.

In each episode $t$, $K$ steps training on $\boldsymbol{u}$ and $\boldsymbol{w}$ are conducted to probe the proper direction of $\boldsymbol{\alpha}$ update. Notably, the parameters of proxy model $\boldsymbol{u}$ and reference model $\boldsymbol{w}$ are synchronized at the beginning of probing, forming a Twin Networks for bi-level DatA mixturE optiMization (TANDEM) framework. Since the updating of $\boldsymbol{\alpha}$ requires $\boldsymbol{u}, \boldsymbol{w}$ probing, in practice, we decrease the frequency of updating $\boldsymbol{\alpha}$ to reduce the computational cost, and leave the proxy model $\boldsymbol{u}$ trained freely for $E$ steps before the next $\boldsymbol{\alpha}$ update (see Figure 1(b)). Altogether the updates of $\boldsymbol{u}, \boldsymbol{w}, \boldsymbol{\alpha}$, the data mixture optimization (DMO) problem can be solved efficiently. The overall computation graph of TANDEM is outlined in Figure 1(b), and a detailed workflow is summarized in Algorithm 1 in Appendix A.

**Convergence Analysis**   Next, we explore the convergence rate of our proposed method. For a non-convex bi-level optimization problem, it is standard to study its first-order stationary convergence result e.g., [30, 19]. Notably, our problem (2) is a constrain problem over $\boldsymbol{\alpha} \in \mathcal{A}$. Thus, it should be considered in first-order stationary condition as in [9, 30]. The convergence result of our method is summarized in the following theorem.

**Theorem 1** (Informally). *For sufficiently large $T$, under mild assumptions (detailed in Appendix A), for $\boldsymbol{\alpha}^{(t)}$ obtained in TANDEM Algorithm 1, it converges to first-order stationary point of problem* (2) *in the rate of $\mathcal{O}(T^{-\frac{1}{4}})$ by properly selecting $\gamma, K, E, \eta_{\boldsymbol{\alpha}}, \eta_{\boldsymbol{w}}$, and $\eta_{\boldsymbol{u}}$.*

The proof is left in Appendix A. Theorem 1 indicates that TANDEM theoretically ensures the optimality of the learned mixture ratio.

## 2.3   TANDEM Improves Existing DMO Methods

We discuss the relationship between TANDEM and two existing methods, DoReMi [37] and DoGE [7]. By comparing the hyper-gradient $\Delta$ [3] that determines $\boldsymbol{\alpha}$ updates in different methods, we show that our TANDEM draws common insights from previous works and improves upon them.

DoReMi updates the mixture ratio according to the excess loss of a proxy model $\boldsymbol{u}$ relative to a reference model $\bar{\boldsymbol{w}}$ trained on uniformly sampled data. DoGE [7] pioneers bi-level optimization to settle data mixtures and tracks the influence [28] of each domain on the validation data. Specifically, $\Delta_{\text{DoGE}} = \left\langle \nabla \mathcal{L}_{\text{train}}\left(\boldsymbol{\alpha}^{(t)}, \boldsymbol{u}^{(t)}\right), \nabla \mathcal{L}_{\text{val}}\left(\boldsymbol{\alpha}^{(t)}, \boldsymbol{u}^{(t)}\right) \right\rangle$. The inner product $\langle \cdot \rangle$ is essentially a first-order approximation of the domain-wise loss difference between a proxy model and a reference model, where the reference model is obtained by one-step update on the validation data.

Outlined in Table 1, we see that $\Delta$ of all three methods takes the form of per-domain loss difference between a proxy model and a reference model. Nevertheless, contrasting DoReMi, which adopts a

---

[3]DoReMi and DoGE utilize exponential gradient descent to update $\boldsymbol{\alpha}$ where $\boldsymbol{\alpha}^{(t+1)} \propto \boldsymbol{\alpha}^{(t)} \exp(-\eta_{\alpha} \Delta)$

Table 1: Summary of $\Delta$ in DoReMi, DoGE and TANDEM. All three methods update $\boldsymbol{\alpha}$ with two models.

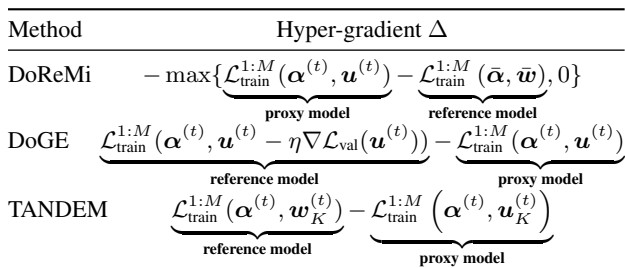

| Method | Hyper-gradient $\Delta$ |
|---|---|
| DoReMi | $-\max\{\underbrace{\mathcal{L}_{\text{train}}^{1:M}(\boldsymbol{\alpha}^{(t)}, \boldsymbol{u}^{(t)})}_{\text{proxy model}} - \underbrace{\mathcal{L}_{\text{train}}^{1:M}(\bar{\boldsymbol{\alpha}}, \bar{\boldsymbol{w}})}_{\text{reference model}}, 0\}$ |
| DoGE | $\underbrace{\mathcal{L}_{\text{train}}^{1:M}(\boldsymbol{\alpha}^{(t)}, \boldsymbol{u}^{(t)} - \eta\nabla\mathcal{L}_{\text{val}}(\boldsymbol{u}^{(t)}))}_{\text{reference model}} - \underbrace{\mathcal{L}_{\text{train}}^{1:M}(\boldsymbol{\alpha}^{(t)}, \boldsymbol{u}^{(t)})}_{\text{proxy model}}$ |
| TANDEM | $\underbrace{\mathcal{L}_{\text{train}}^{1:M}(\boldsymbol{\alpha}^{(t)}, \boldsymbol{w}_K^{(t)})}_{\text{reference model}} - \underbrace{\mathcal{L}_{\text{train}}^{1:M}(\boldsymbol{\alpha}^{(t)}, \boldsymbol{u}_K^{(t)})}_{\text{proxy model}}$ |

Figure 2: SFT exhibits lower gradient alignment than pretraining

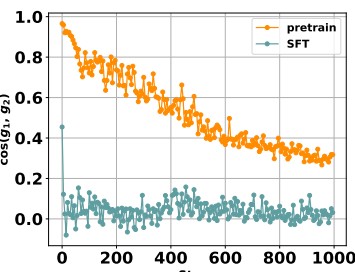

fixed reference model, the reference model $\boldsymbol{w}$ in TANDEM is dynamic, which better captures the current training status. In its original form, DoGE incurs significant memory overhead as it maintains the per-domain gradients. Relaying explicitly on the gradient estimation also makes it vulnerable to instability issues in high gradient variance scenarios like supervised fine-tuning (SFT). Our TANDEM better adapts, as increasing the probing step K helps reduce the variance on $\Delta$, as will be elaborated in Section 4.3. In Figure 2, we show that SFT exhibits higher gradient variance than pretraining. The alignment of $g_1$ and $g_2$ $\cos(g_1, g_2)$ is used as a proxy of the gradient variance, where $g_1$ and $g_2$ are gradients evaluated on different batches. Lower alignment $\cos(g_1, g_2)$ indicates larger gradient variance.

## 2.4 Domain Reweighting Beyond Traditional Settings

In the above, we discuss our method to determine the data mixture ratio by solving bi-level problem (1). The circumstances under which DMO is most effective remain to be explored. Theoretically, the standard training setting sets $\boldsymbol{\alpha}_m = 1/M$ for every $m$. Let us check the following proposition.

**Proposition 1.** *Assume $\mathcal{L}_{train}^m = \mathcal{L}_{val}^m$, the uniform data mixture ratio $\bar{\alpha}_m = \frac{1}{M}$ for $m = 1, \ldots, M$ constitutes a valid solution of the bi-level mixture optimization* (1).

The proof is left in Appendix B. As can be seen, when the generalization gap $|\mathcal{L}_{\text{val}}^m - \mathcal{L}_{\text{train}}^m|$ approaches zero, uniform weighting emerges as a valid solution, thus making the reweighting less significant. This holds for the conventional data-abundant scenario. When $\bar{\alpha}_m = \frac{1}{M}$, the train data and validation data are independently identically distributed, so the loss gap is approximately zero for the first epoch training. [4]. This aligns with the empirical observations that uniform weighting is highly competitive that many DMO methods can not consistently outperform [2].

When there is limited data in specific domains or the trained model overfits to some domain, since both phenomenons result in a large generalization gap, the reweighting technique over data domains becomes significant. In fact, even though LLMs are trained on massive datasets overall, it is common for specific domains to be relatively small, e.g., specialized scientific literature, low-resource languages, or domain-specific user interactions. In practice, the data-restricted scenario is ubiquitous. Furthermore, in SFT, repeated passes over the same domain data exacerbate overfitting and widen the generalization gap [10], which presents more interesting opportunities for domain reweighting. Please refer to Appendix F.1 for empirical evidence.

## 3 Related Work

Data mixture optimization is drawing increasing attention for designing LLMs with comprehensive and balanced capabilities. Our work follows the recent trend of formulating the DMO as a bi-level optimization. We summarize the related literature from these two strands of research in the following.

**Data Mixture Optimization** Conventional industry practices determine the optimal cross-domain composition with human expertise [6, 34, 11]. To circumvent the exhaustive trial-and-error, various heuristics has been explored. DoReMi [37] settles the mixture ratio by

---

[4]In the data-abundant training scenario, samples are consumed only once.

minimizing the worst-domain excess loss relative to a well-trained reference model, pursuing good performance in all domains. [23, 41, 16] fit global data mixing laws, predict the performance on other mixtures, and search for those with good performances. Other works capture the inter-domain relations through domain embeddings [39] or training several models [3].

Skill-It [3] then utilize the inter-domain relations to establish a local data mixing law that predicts the per-domain loss. Aioli [2] improves Skill-It by dynamically updating the inter-domain relation matrix according to current model states. Nevertheless, these approaches remain theoretically ungrounded due to their inductive nature, and incorporating the fitting of the mixing law incurs additional approximation error. DOGE [7] pioneers bi-level optimization to settle data mixtures and tracks the influence [28] of each domain on the validation set. However, assessing the influence relies directly on the per-domain gradient estimation, which undermines its efficacy in high gradient variance scenar-

Table 2: Computational complexity of different DMO methods.

| Method | Computational Complexity |
|---|---|
| Vanilla Train | $CTE$ |
| DoReMi | $\frac{7}{3}CTE$ |
| DoGE | $2CTE$ |
| Skill-it | $(\frac{M}{4}+1)CTE + \frac{M}{3}CT$ |
| Aioli | $CTE + MCTK + \frac{1}{3}M^2CT$ |
| TANDEM | $CTE + 2CTK + \frac{2}{3}CT$ |

ios while incurring large memory overhead. Comparison of the computational complexity of these strategies is given in Table 2, where $C$ is the complexity of one step training of a model. $T$ is the update number of $\boldsymbol{\alpha}$. Typically, the overall computational cost of these methods is of the order: Aioli > Skill-it > DoReMi ≈ DoGE ≈ TANDEM [5] [6]. Note that, although the complexity of DoGE is not particularly large, it requires per-domain gradient computation, which is unfriendly to the computing kernel and takes significantly more time in practice.

**Bi-level Optimization**   Bi-level optimization has been an important research topic in many scientific disciplines, however, solving the bi-level optimization problem is challenging due to the complicated dependency of the upper-level and lower-level problems. Typical bi-level optimization algorithms [24, 5, 29, 12, 4, 15] requires estimation of the implicit gradient, which requires second-order derivatives on the lower-level variables. Incorporating the Hessian makes it computationally prohibitive for large-scale problems. Recently [30, 19, 20] pioneer the penalized methods, where the inner-level problem is reformulated into an penalty. As first-order gradient-based approaches, the penalized methods avoid the estimation of the Hessian or the Jacobian. Though effective in theory, their practical applications in large-scale LLM settings remain rarely explored. The most similar to ours is the recent ScaleBio [27], which belongs to this category, and is specially tailored for large-scale data mixture optimization problems. Nevertheless, the solving procedure is different, TANDEM enjoys more stable training by synchronizing $\boldsymbol{u}$ and $\boldsymbol{w}$ periodically.

## 4   Experiments

In this section, we compare TANDEM to state-of-the-art algorithms in Section 4.2. Then we analyse the effectiveness of each design ingredient in Section 4.3.

### 4.1   Experimental Setup

We consider three application scenarios: conventional data-abundant pretraining, data-restricted training, and supervised fine-tuning. A brief summarization of the experimental setup is introduced below, while complete hyper-parameter settings and implementation details are in Appendix C.

**Data-Abundant Scenario:**   For the data-aboundant scenario, we train 160M GPT-style LMs [1] on a 6B sampled version of SlimPajama [31] as in [2]. SlimPajama consists of 7 domains: ArXiv, Books, CommonCrawl, C4, Github, StackExchange, and Wikipedia. The statistics of this sampled corpus are given in Figure 4. We set $E = 20$, $K = 5$, train with batch size 8 and context length 2048 for 40000 steps (with respect to updates of proxy model $\boldsymbol{u}$, so the mixture ratio $\boldsymbol{\alpha}$ is updated for ∼2000 steps.) as [2]. Though the SlimPajama-6B corpus exhibits significant domain imbalance, 40K

---

[5]We assume backward takes 2× computation as the forward process.

[6]Skill-it trains $M$ models with data from each domain for $H$ steps, we use $H = \frac{1}{4}TE$ in our analysis.

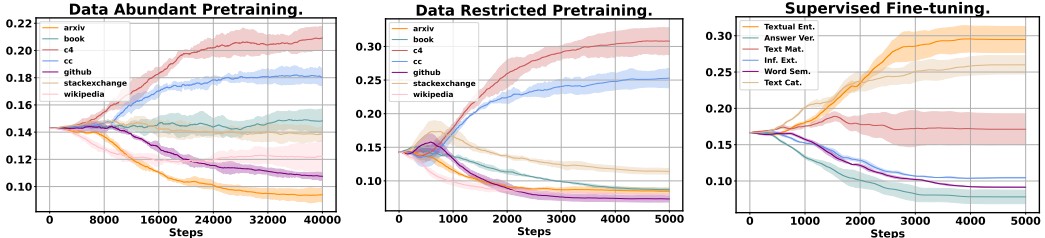

Figure 3: Step-wise data mixture ratio evolution under three scenarios. We repeat the DMO 3 times and the 95% confidence intervals are given. (a) data-abundant pretraining (b) data-restricted pretraining and (c) supervised fine-tuning.

steps of training doesn't deplete even the smallest domain, so this setting constitutes a data-abundant one-epoch scenario. The penalty constant $\gamma$ is set to 1 across all the experiments. Experiments are conducted on eight NVIDIA Hopper H-100s.

**Data-Restricted Scenario:** For the data-restricted scenario, we construct a 300M sampled version of SlimPajama and keep the domain distribution unchanged. GPT-style LMs of size 160M, 410M and 1B are trained to examine the scalability of TANDEM. In this scenario, we train with $K = E = 5$, batch size 128, and context length 512 for 5000 steps, which ensures that samples in small domains like Arxiv, Books, StackExchange, and Wikipedia are exposed more than once. After DMO, the learned mixture ratio is utilized to resample the 300M corpus for the final model training.

**Supervised Fine-tuning:** For supervised fine-tuning, we use 6 major categories (containing 99 tasks) from Natural Instructions [25, 35], Textual Entailment, Answer Verification, Text Matching, Information Extraction, Word Extraction and Text Categorization. Prioritizing diversity of capabilities and formats, this corpus is comprised of open-ended text generation, multiple choice, and True/False tasks. Due to the limited space, the statistics are given in Appendix D. In this scenario, we train a pretrained Qwen2-500M model [40] with $K = 10$, $E = 10$, batch size 64, and context length 512 for 5000 steps. Different from pretraining where the majority of samples are only trained once, in instruction tuning, each sample is on average exposed 1.15 times.

## 4.2 Comparisons with State of the Arts

We compare TANDEM against various state-of-the-art methods. **Uniform** A simple baseline that uniformly mixes groups and requires zero extra training runs. **DoReMi** [37] adopts the idea of distributionally robust optimization and searches for the data mixture ratio by minimizing the worst-domain excess loss over a reference model trained with the uniform strategy. **DOGE** [7] solves the DMO problem by tracking the data influence of each domain on the validation set and up-weights the most influential domains. **Skill-It** [3] trains several models to fit an inter-domain relation matrix, which is later used to establish an incremental data mixing law that induces a mixture ratio $\alpha$ update rule. **Aioli** [2] improves Skill-It by dynamically updating the inter-domain relation matrix according to current model states. For the baselines, We use the published implementations. [7] Averaged results from 3 runs are reported. Due to limited space, the standard deviation is given in Appendix E

**Data-Abundant Pretraining** In Table 3 (Upper), we see that TANDEM discovers mixture ratios with comparative performance. As discussed in Section 2.4, in this scenario, the uniform strategy is highly competitive. The mixture ratio for the baselines is obtained from Aioli [2]. We show the step-wise data mixture ratio evolution of TANDEM during DMO in Figure 3 (Left). Note that the uniform strategy is not the only valid solution to the DMO problem, and TANDEM finds another solution that performs equally well. The detailed learned mixture ratio of each method is given in Appendix G).

---

[7]**DoReMi**: https://github.com/sangmichaelxie/doremi, **DOGE**: https://github.com/Olivia-fsm/DoGE, **Skill-It** and **Aioli**: https://github.com/HazyResearch/aioli

Table 3: Comparison for the 160M GPT-style model on SlimPajama in the data-abundant scenario (Upper) and data-restricted scenario (Lower). Per-domain perplexity is reported and "Average" represents the exponential of the average loss across all domains as in [7, 2]. † denotes the results reported use the mixture ratio given in Aioli [2].

| | Methods | Arxiv | Book | C4 | CommonCrawl | Github | Stackexchange | Wikipedia | Average |
|---|---|---|---|---|---|---|---|---|---|
| Data Abundant Regime | Uniform | **11.46** | 62.53 | 66.43 | 59.29 | 6.71 | 13.98 | 28.31 | 25.74 |
| | DoReMi† | 12.71 | 80.09 | 82.60 | 71.76 | 5.75 | 14.26 | 29.54 | 28.32 |
| | DoGE† | 12.89 | **51.50** | **54.32** | **49.34** | 8.48 | 16.77 | 37.21 | 26.60 |
| | Skill-It† | 11.76 | 62.24 | 64.58 | 59.84 | **6.36** | **12.36** | 34.87 | 25.87 |
| | Aioli† | 11.47 | 61.89 | 65.52 | 58.24 | 6.74 | 14.08 | 28.48 | 25.66 |
| | TANDEM | 11.53 | 61.82 | 65.92 | 58.86 | 6.63 | 13.76 | **27.26** | **25.43** |
| Data Restricted Regime | Uniform | 18.05 | 65.86 | 71.05 | 63.76 | 9.37 | 17.94 | 34.27 | 31.53 |
| | DoReMi | 18.90 | 80.29 | 89.05 | 79.02 | 10.24 | 19.74 | 43.20 | 36.91 |
| | DoGE | 17.76 | 60.88 | 65.08 | 58.81 | 9.00 | 17.71 | 33.94 | 30.10 |
| | Skill-It | 20.93 | **52.00** | 57.11 | **49.50** | **8.77** | **16.74** | 40.49 | 29.24 |
| | Aioli | 17.68 | 62.48 | 69.02 | 61.44 | 9.26 | 17.79 | 33.06 | 30.67 |
| | TANDEM | **16.85** | 52.75 | **56.82** | 51.11 | 8.99 | 18.21 | **32.52** | **28.07** |

Figure 4: SlimPajama-6B Statistics.

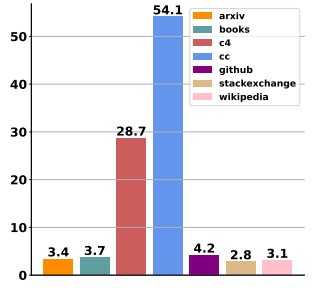

Table 4: Comparison for models of different sizes.

| | 160M | | 410M | | 1B | |
|---|---|---|---|---|---|---|
| | Avg. | Time | Avg. | Time | Avg. | Time |
| Uniform | 31.53 | - | 29.59 | - | 29.91 | - |
| DoReMi | 36.91 | 16$_{min}$ | 54.61 | 31$_{min}$ | 56.53 | 41$_{min}$ |
| DoGE | 30.10 | 65$_{min}$ | 27.45 | 172$_{min}$ | - | - |
| Skill-It | 29.24 | 28$_{min}$ | 27.70 | 59$_{min}$ | 27.15 | 80$_{min}$ |
| Aioli | 31.19 | 36$_{min}$ | 28.79 | 64$_{min}$ | 28.07 | 93$_{min}$ |
| TANDEM | **28.07** | 26$_{min}$ | **25.00** | 57$_{min}$ | **24.35** | 77$_{min}$ |

**Data-Restricted Pretraining** For the data-restricted training scenario, TANDEM significantly outperforms baselines as shown in Table 3 (Lower). For instance, it achieves 28.07 averaged perplexity, surpassing the most competitive Skill-It by 1.17 and the uniform baseline by 3.46. In this scenario, the uniform strategy is no longer competitive. Equally assigning $\frac{1}{M}$ weights potentially leads to overfitting in small domains (repeated multiple times), while leaving the large domains underfitting. The mixture ratios learned with TANDEM and other baselines are shown in Figure 5, and the step-wise data mixture ratio evolution during the data mixture optimization is given in Figure 3 (Middle). We see that after DMO, CommonCrawl and C4 take the majority, driven by their extensive lexical diversity and complex semantics. Nevertheless, compared to the original data distribution, TANDEM up-weights the small domains Arxiv, Books, Github, StackExchange and Wikipedia from 3.4%, 3.7%, 4.2%, 2.8%, 3.1% to 8.9%, 8.5%, 7.6%, 11.5%, 7.9% respectively, preventing these small domains from being overwhelmed while avoiding potential overfitting. Besides, we inspect the scalability of TANDEM with three different scales (160M, 410M, 1B) in Table 4. TANDEM consistently outperforms the baselines with a large margin while not incurs much computational overhead. We omit DoGE for the 1B model experiment due to its large memory consumption.

**Supervised Fine-tuning** The results for supervised fine-tuning are given in Table 5. The overall averaged accuracy as well as the test loss are reported. From Table 5, TANDEM outperforms or is on par with the other data mixture optimization methods in 5 out of 6 task clusters. Showing its effectiveness in the supervised fine-tuning scenario. Besides, we consider a more fine-grained task-level SFT (instead of the task cluster) case, please refer to Appendix D.

### 4.3 Analyses

To evaluate the effectiveness of each design component, we conduct ablation experiments. The model used is the 160M GPT-style model unless specified. Due to limited space, we focus on the effect of synchronizing $u$ and $w$, the effect of the probing steps $K$, and the robustness of TANDEM to model

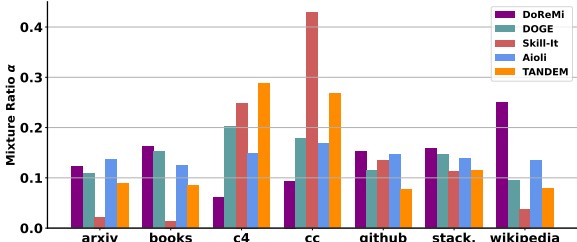
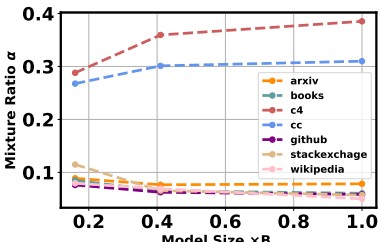

Figure 5: Mixture ratio learned by different methods. Figure 6: Impact of different model sizes.

Table 5: Comparison for the 500M Qwen-2 model on a mixture of 6 major categories (99 tasks in total) of supervised fine-tuning tasks.

| Method | Textual Ent. | Answer Ver. | Text Mat. | Inf. Ext. | Word Sem. | Text Cat. | Test Loss ↓ | Avg. Metric ↑ |
|--------|-------------|-------------|-----------|-----------|-----------|-----------|-------------|---------------|
| Uniform | 84.4 | 75.1 | 86.2 | 77.8 | 88.3 | 83.2 | 0.231 | 82.5 |
| DoReMi | 85.3 | 74.9 | 83.6 | 78.1 | 87.0 | 79.0 | 0.249 | 81.6 |
| DoGE | 81.4 | 76.7 | **86.5** | **78.6** | 88.3 | 82.4 | 0.297 | 82.4 |
| Skill-It | 84.1 | 75.0 | 86.4 | 78.3 | 87.9 | 83.7 | 0.232 | 82.6 |
| Aioli | 83.9 | 75.8 | 86.2 | 78.0 | 88.3 | 84.0 | 0.229 | 82.7 |
| TANDEM | **85.3** | **76.3** | 86.2 | **78.6** | **88.5** | **84.9** | **0.208** | **83.3** |

scales. For a more comprehensive analysis, please refer to Appendix F, including the effectiveness of DMO in the real-world large-scale data (past chinchilla [14]) scenario, sensitivity of the final result on $K$ and $E$, the performance of the DMO pretrained model on downstream tasks, as well as the performance of the proxy model $u$.

**The Effect of Synchronizing $u$ and $w$** One obvious characteristic of our method is that the proxy model $u$ and the reference model $w$ are synchronized by setting $w_0^{(t)} = u_0^{(t)}$. We show the effect of the synchronization by inspecting $\mathrm{Dist}(u, w) = \|u - w\|_2$ along the DMO training process. From Figure 7(a), we see that with synchronization, the distance between $u$ and $w$ is well controlled under $1.5e^{-4}$ and gradually contracts during the whole DMO process. This contraction is critical for $w$ and $\alpha$ in the penalized problem (2) to converge to the original bi-level optimization problem (1) as discussed in Section 2. On the contrary, independently maintaining the proxy model $u$ and the reference model $w$ incurs blown-up distance $\mathrm{Dist}(u, w)$ as shown in Figure 7(b).

**The Effect of the Probing Steps K** During DMO, the hyper-gradient $\Delta$ determines the update of $\alpha$. To validate the effectiveness of K in reducing the variance of $\Delta$, we trace $\cos\left(\Delta, \tilde{\Delta}\right)$ through the training. $\tilde{\Delta}$ is the hyper-gradient evaluated using another batch of data other than that of $\Delta$, so $\cos\left(\Delta, \tilde{\Delta}\right)$ serves as a proxy of the variance, the better $\Delta$ and $\tilde{\Delta}$ aligns, the smaller the variance of hyper-gradient is. We inspect under the SFT setting (with the 500M Qwen model) where the gradients exhibited large variance. During the training, $\alpha$ is fixed to prevent the interference of inaccurate mixture ratio.

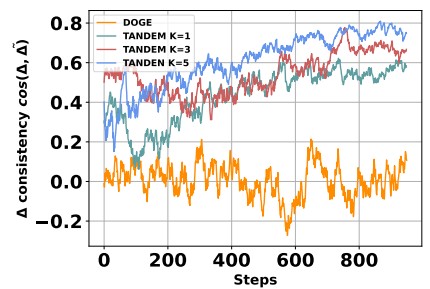

Figure 8: The variance of $\Delta$ decreases with more probing steps.

From Figure 8, we see that DoGE exhibits the largest $\Delta$ variance as it explicitly depends on the noisy parameter gradient estimation. As K increases, the variance of $\Delta$ decrease, leading to more reliable updates of the mixture ratio. Nevertheless, large K increases the computational cost. So in practice and we need to deliberately choose the proper K.

**The Effect of the Model Scale** To investigate how the model size will impact the final mixture ratio. In Figure 6, we compare $\alpha$ learned with models of size 160M, 410M, and 1B. We see that learned

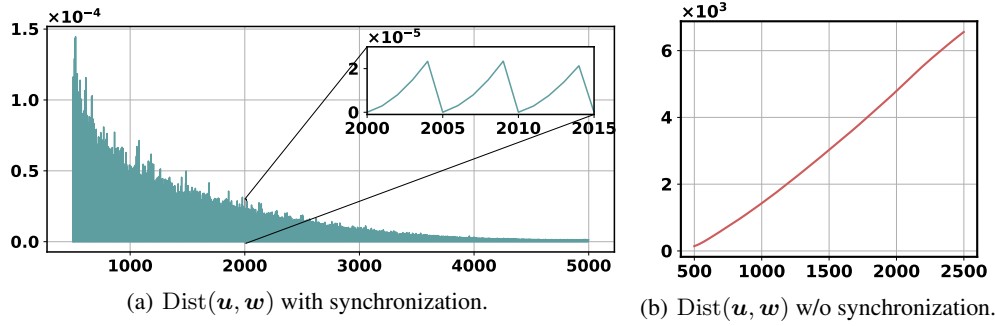

(a) $\mathrm{Dist}(\boldsymbol{u}, \boldsymbol{w})$ with synchronization.

(b) $\mathrm{Dist}(\boldsymbol{u}, \boldsymbol{w})$ w/o synchronization.

Figure 7: The $\mathrm{Dist}(\boldsymbol{u}, \boldsymbol{w})$ evolution comparison during DMO with and without $\boldsymbol{u}, \boldsymbol{w}$ synchronization.

$\boldsymbol{\alpha}$ with larger models are slightly "sharper" than the smaller ones. More specifically, the 1B model further increases the weights of the already large CommonCrawl and C4 while down-weights the others. For large models, due to the increasing capability of memorizing samples, smaller domains are less likely to be overwhelmed, while the risk of potential overfitting increases. The capability of capturing this subtle difference further demonstrates the effectiveness of TANDEM. Nevertheless, the optimal mixture ratios under different model scales share the same trend, so a smaller model can work as a valid proxy for efficient searching.

## 5 Conclusion

In this paper, we propose TANDEM, a principled, efficient and versatile data mixture optimization framework. By solving the DMO bi-level optimization problem, TANDEM ensures the optimality of the learned mixture ratios, along with a $\mathcal{O}(T^{-\frac{1}{4}})$ convergence rate. Besides the algorithmic contribution, from the bi-level optimization perspective, we further demonstrate the limitation of the conventional data-abundant setting in DMO and advocate new settings like data-restricted scenario as well as supervised fine-tuning. Extensive experiments and analysis are conducted to demonstrate the effectiveness of our approach. Our work deepens the understanding of data mixture optimization and expands its application scenarios.

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

# A   Convergence of Bi-level Data Mixture Optimization

## A.1   The Proposed Algorithm

As mentioned in the main part of this paper, the data mixture optimization problem is bi-level. Here we present a theoretical analysis of the proposed TANDEM. The original bi-level optimization problem

$$\mathcal{P}: \quad \min_{\boldsymbol{\alpha} \in \mathcal{A}} \mathcal{L}_{\text{val}}\left(\boldsymbol{w}^*(\boldsymbol{\alpha})\right) \quad \text{s.t. } \boldsymbol{w}^*(\boldsymbol{\alpha}) \in S^*(\boldsymbol{\alpha}) := \arg\min_{\boldsymbol{w}} \mathcal{L}_{\text{train}}(\boldsymbol{\alpha}, \boldsymbol{w}) \tag{6}$$

is difficult to solve owing to the imposed hard constraint. We turn to the Lagrangian problem of $\mathcal{P}$ such that

$$\mathcal{P}_\gamma: \quad \min_{\boldsymbol{\alpha} \in \mathcal{A}, \boldsymbol{w}} \mathcal{L}_{\text{val}}\left(\boldsymbol{w}\right) + \gamma \left( \mathcal{L}_{\text{train}}(\boldsymbol{\alpha}, \boldsymbol{w}) - \min_{\boldsymbol{w}} \mathcal{L}_{\text{train}}(\boldsymbol{\alpha}, \boldsymbol{w}) \right) \tag{7}$$

There is a vast variety of existing literature, e.g. [30, 19, 36] discuss the relationship between the Lagrangian problem $\mathcal{P}_\gamma$ and the original problem $\mathcal{P}$. In a word, when taking $\gamma$ sufficient large, the solution of $\mathcal{P}_\gamma$ will approximate the solution of $\mathcal{P}$. Thereafter, we can develop the algorithm to $\mathcal{P}_\gamma$ to solve $\mathcal{P}$, as we did in this paper.

---

**Algorithm 1** Twin Networks for bi-level DatA mixturE optiMization (TANDEM)

---

**Input:**
    Train set $\mathcal{D}_{\text{train}}$, validation set $\mathcal{D}_{\text{val}}$ comprised of M domains.
    Episode number $T$, Episode length $E$, Probing length for each episode $K$,
    Learning rate $\eta_{\boldsymbol{w}}, \eta_{\boldsymbol{u}}, \eta_{\boldsymbol{\alpha}}$ for $\boldsymbol{w}$ (reference), $\boldsymbol{u}$ (proxy) and $\boldsymbol{\alpha}$ (mixture) respectively.
1: Initialize proxy model parameters $\boldsymbol{u}^0$ and domain weights $\boldsymbol{\alpha}^0$.
2: **for** $t = 0$ **to** $T - 1$ **do**

    *// Mixture ratio $\boldsymbol{\alpha}$ update.*
3:    Set $\boldsymbol{w}_0^{(t)}, \boldsymbol{u}_0^{(t)} \leftarrow \boldsymbol{u}^{(t)}$.
4:    **for** $k = 0$ **to** $K - 1$ **do**
5:      $\boldsymbol{u}_{k+1}^{(t)} = \boldsymbol{u}_k^{(t)} - \eta_{\boldsymbol{u}} \nabla_{\boldsymbol{u}} \mathcal{L}_{\text{train}}(\boldsymbol{\alpha}^{(t)}, \boldsymbol{u}_k^{(t)})$.
6:      $\boldsymbol{w}_{k+1}^{(t)} = \boldsymbol{w}_k^{(t)} - \eta_{\boldsymbol{w}} (\nabla_{\boldsymbol{w}} \mathcal{L}_{\text{val}}(\boldsymbol{w}_k^{(t)}) + \gamma \nabla_{\boldsymbol{w}} \mathcal{L}_{\text{train}}(\boldsymbol{\alpha}^{(t)}, \boldsymbol{w}_k^{(t)}))$.
7:    **end for**
8:    $\boldsymbol{\alpha}^{(t+1)} = \Pi_{\mathcal{A}}(\boldsymbol{\alpha}^{(t)} - \eta_{\boldsymbol{\alpha}} (\underbrace{\mathcal{L}_{\text{train}}^{1:M}(\boldsymbol{\alpha}^{(t)}, \boldsymbol{w}_K^{(t)})}_{\text{reference model}} - \underbrace{\mathcal{L}_{\text{train}}^{1:M}(\boldsymbol{\alpha}^{(t)}, \boldsymbol{u}_K^{(t)})}_{\text{proxy model}}))$

    *// Free model weights $\boldsymbol{u}$ update.*
9:    **for** $e = 0$ **to** $E - 1$ **do**
10:      $\boldsymbol{u}_{e+1}^{(t)} = \boldsymbol{u}_e^{(t)} - \eta_{\boldsymbol{u}} \nabla_{\boldsymbol{u}} \mathcal{L}_{\text{train}}(\boldsymbol{\alpha}^{(t+1)}, \boldsymbol{u}_e^{(t)})$.
11:    **end for**
12:    Set $\boldsymbol{u}^{(t+1)} \leftarrow \boldsymbol{u}_E^{(t)}$.
13: **end for**
**Output:**
    domain weights $\boldsymbol{\alpha}^{(T)}$.

---

A full workflow of Twin Networks for bi-level DatA mixturE optiMizatio (TANDEM) is shown in Algorithm1. TANDEM alternates between updating the mixture ratio $\boldsymbol{\alpha}$ and the proxy model $\boldsymbol{u}$, whereas $\boldsymbol{u}$ is used to approximate the minimum of $\mathcal{L}_{\text{train}}(\boldsymbol{\alpha}, \boldsymbol{w})$. Update of the mixture ratio $\boldsymbol{\alpha}$ requires probing the data efficacy of each domain. During probing, we optimize the reference model $\boldsymbol{w}$, as well as the proxy model $\boldsymbol{u}$ for $K$ steps. $\boldsymbol{w}$ and $\boldsymbol{u}$ are respectively trained on the train set and the train set plus validation set, their incurred loss gap captures the gain of incorporating the additional validation data. TANDEM then up-weights domains that benefit more from additional data. Notably, the proxy model $\boldsymbol{u}$ and reference model $\boldsymbol{w}$ are synchronized at the beginning of probing. Since the updating of $\boldsymbol{\alpha}$ requires $\boldsymbol{u}, \boldsymbol{w}$ probing, we decrease the frequency of updating $\boldsymbol{\alpha}$ to reduce the computational cost, and leave the $\boldsymbol{u}$ trained freely for $E$ steps before the next $\boldsymbol{\alpha}$ update.

## A.2   Convergence Rate

Next, we explore the convergence rate of the proposed Algorithm 1. For the original problem $\mathcal{P}$ (6), its convergence property under the iterates obtained by solving the Lagrange problem (7) has been

explored, under different regularity conditions [30, 19]. In this paper, we will present our results under a mild Polyak-Łojasiewicz (PL) condition [43, 17, 30, 44, 26, 42], which has been widely imposed to study the bi-level optimization problem. Technically, we will impose the following Assumptions to derive the convergence rate. Before illustrating our Assumptions, we need the following definitions to simplify the notations. We denote

$$\mathcal{H}_\gamma(\boldsymbol{\alpha}, \boldsymbol{w}) = \mathcal{L}_{\text{val}}(\boldsymbol{w}) + \gamma \left( \mathcal{L}_{\text{train}}(\boldsymbol{\alpha}, \boldsymbol{w}) - \min_{\boldsymbol{w}} \mathcal{L}_{\text{train}}(\boldsymbol{\alpha}, \boldsymbol{w}) \right), \tag{8}$$

with $S_\gamma^*(\boldsymbol{\alpha}) = \{\boldsymbol{w} : \boldsymbol{w} \in \arg\min_{\boldsymbol{w}} \mathcal{H}_\gamma(\boldsymbol{\alpha}, \boldsymbol{w})\}$, and $S^*(\boldsymbol{\alpha}) = \{\boldsymbol{w} : \boldsymbol{w} \in \arg\min_{\boldsymbol{w}} \mathcal{L}_{\text{train}}(\boldsymbol{\alpha}, \boldsymbol{w})\}$. We further define

$$\mathcal{H}(\boldsymbol{\alpha}) = \inf_{\boldsymbol{w} \in S^*(\boldsymbol{\alpha})} \mathcal{L}_{\text{val}}(\boldsymbol{w}), \tag{9}$$

where $\mathcal{H}(\boldsymbol{\alpha})$ is well-defined since $\mathcal{L}_{\text{train}}(\boldsymbol{\alpha}, \boldsymbol{w})$ is continuous to $\boldsymbol{w}$. Besides that, the minimum of $\mathcal{H}(\boldsymbol{\alpha})$ is exactly the solution to original problem $\mathcal{P}$ (6).

**Assumption 1** (PL condition). *For any $\boldsymbol{\alpha} \in \mathcal{A}$, both $\mathcal{L}_{\text{train}}(\boldsymbol{\alpha}, \boldsymbol{w})$ and $\mathcal{H}_\gamma(\boldsymbol{\alpha}, \boldsymbol{w})$ satisfy the PL inequality with coefficient $\mu$ and $\mu_\gamma$, respectively. That says:*

$$\mathcal{L}_{\text{train}}(\boldsymbol{\alpha}, \boldsymbol{w}) - \min_{\boldsymbol{w}} \mathcal{L}_{\text{train}}(\boldsymbol{\alpha}, \boldsymbol{w}) \leq \frac{1}{2\mu} \|\nabla_{\boldsymbol{w}} \mathcal{L}_{\text{train}}(\boldsymbol{\alpha}, \boldsymbol{w})\|^2 \tag{10}$$

*and*

$$\mathcal{H}_\gamma(\boldsymbol{\alpha}, \boldsymbol{w}) - \min_{\boldsymbol{w}} \mathcal{H}_\gamma(\boldsymbol{\alpha}, \boldsymbol{w}) \leq \frac{1}{2\mu_\gamma} \|\nabla \mathcal{H}_\gamma(\boldsymbol{\alpha}, \boldsymbol{w})\|^2. \tag{11}$$

*Moreover, the coefficient $\mu_\gamma$ satisfies $\lim_{\gamma \to \infty} \frac{\mu_\gamma}{\gamma} = 1$.*

**Assumption 2** (Smoothness). *For any $\boldsymbol{\alpha} \in \mathcal{A}$,*

1. *Both $\nabla_{\boldsymbol{w}} \mathcal{L}_{\text{train}}(\boldsymbol{\alpha}, \boldsymbol{w})$ and $\nabla_{\boldsymbol{w}} \mathcal{L}_{\text{val}}(\boldsymbol{w})$ are Lipschitz continuous to $\boldsymbol{\alpha}$ (hold for $\nabla_{\boldsymbol{w}} \mathcal{L}_{\text{train}}(\boldsymbol{\alpha}, \boldsymbol{w})$) and $\boldsymbol{w}$ on coefficient $L$.*

2. *For any $\boldsymbol{\alpha} \in \mathcal{A}$, both $\mathcal{L}_{\text{train}}(\boldsymbol{\alpha}, \boldsymbol{w})$ and $\mathcal{L}_{\text{val}}(\boldsymbol{w})$ are Lipschitz continuous to $\boldsymbol{w}$ with coefficient $B$.*

**Assumption 3** (Bounded Hessian). *For any $\boldsymbol{\alpha} \in \mathcal{A}$, $\boldsymbol{w} \in S^*(\boldsymbol{\alpha})$, there exists positive constants $\lambda, \rho$, satisfying Hessian matrices $\nabla_{\boldsymbol{ww}} \mathcal{L}_{\text{train}}(\boldsymbol{\alpha}, \boldsymbol{w}) \succeq \lambda^8$ and $\nabla_{\boldsymbol{\alpha w}}^2 \mathcal{L}_{\text{train}}(\boldsymbol{\alpha}, \boldsymbol{w}) \preceq \rho$.*

**Assumption 4** (Lipschitz Hessian). *For any $\boldsymbol{\alpha} \in \mathcal{A}$, $\mathcal{L}_{\text{train}}(\boldsymbol{\alpha}, \boldsymbol{w})$ is twice-times continuous differentiable, and the Hessian matrices $\nabla_{\boldsymbol{\alpha w}}^2 \mathcal{L}_{\text{train}}(\boldsymbol{\alpha}, \boldsymbol{w})$, $\nabla_{\boldsymbol{ww}}^2 \mathcal{L}_{\text{train}}(\boldsymbol{\alpha}, \boldsymbol{w})$ are all Lipschitz continuous to $\boldsymbol{w}$ with coefficient $H$.*

**Assumption 5** (Bounded Loss Function). *The non-negative loss function $\mathcal{L}_{\text{train}}(\boldsymbol{\alpha}, \boldsymbol{w})$, $\mathcal{L}_{\text{val}}(\boldsymbol{w})$ are uniformly bounded by positive constant $D$.*

Notably, for the bounded Hessian condition, due to the structure of $\mathcal{L}_{\text{train}}(\boldsymbol{\alpha}, \boldsymbol{w})$, it can be implied by the lower bounded $\nabla_{\boldsymbol{ww}} \mathcal{L}_{\text{train}}^m(\boldsymbol{w})$ and an upper bounded $\nabla_{\boldsymbol{w}} \mathcal{L}_{\text{train}}(\boldsymbol{w})$ for any $m$. Moreover, it worth noting that for bi-level optimization problem, it is standard to impose some regularity conditions to the Hessian matrix. For examples, the smooth Hessian in [30, 19] and lower bounded Hessian in [19]. Therefore, the imposed bounded Hessian Assumption 3 can be considered as a mild condition.

Next, we present a lemma to characterize the gap between the gradient of $\nabla_{\boldsymbol{\alpha}} \mathcal{H}(\boldsymbol{\alpha})$ and $\nabla_{\boldsymbol{\alpha}} \mathcal{H}_\gamma(\boldsymbol{\alpha}, \boldsymbol{w})$.

**Lemma 1.** *Under Assumptions 1-4, for any given $\boldsymbol{\alpha}$ and $\boldsymbol{w}_\gamma^*(\boldsymbol{\alpha}) \in S_\gamma^*(\boldsymbol{\alpha})$, it holds*

$$\left\| \nabla_{\boldsymbol{\alpha}} \mathcal{H}(\boldsymbol{\alpha}) - \frac{\partial}{\partial \boldsymbol{\alpha}} \mathcal{H}_\gamma(\boldsymbol{\alpha}, \boldsymbol{w}_\gamma^*(\boldsymbol{\alpha})) \right\| \leq \frac{1}{\gamma} \left( \frac{HB^2}{\mu^2} \left( \frac{\rho}{\lambda} + 1 \right) + \frac{\rho LB}{\mu \lambda} \right) \tag{12}$$

*Proof.* Without loss of generality, suppose that $\mathcal{H}(\boldsymbol{\alpha}) = \mathcal{L}_{\text{val}}(\boldsymbol{w}^*(\boldsymbol{\alpha}))$, due to the chain rule, we have

$$\nabla_{\boldsymbol{\alpha}} \mathcal{H}(\boldsymbol{\alpha}) = \nabla_{\boldsymbol{\alpha}} \mathcal{L}_{\text{val}}(\boldsymbol{w}^*(\boldsymbol{\alpha})) = \nabla_{\boldsymbol{\alpha}} \boldsymbol{w}^*(\boldsymbol{\alpha})^\top \nabla_{\boldsymbol{w}} \mathcal{L}_{\text{val}}(\boldsymbol{w}^*(\boldsymbol{\alpha})). \tag{13}$$

---

[8]For two matrices $\boldsymbol{A}$, $\boldsymbol{A} \succeq \lambda$ means $\boldsymbol{A} - \lambda \boldsymbol{I}$ is a positively semi-definite matrix.

To simplify the notation, we denote $\boldsymbol{w}_\gamma^*(\boldsymbol{\alpha})$ and $\boldsymbol{w}^*(\boldsymbol{\alpha})$ as $\boldsymbol{w}_\gamma^*$ and $\boldsymbol{w}^*$ in the sequel. For the penalized problem, given any $\boldsymbol{w}_\gamma^*(\boldsymbol{\alpha})$, we have

$$
\begin{aligned}
&\frac{\partial}{\partial \boldsymbol{\alpha}} \mathcal{H}_\gamma(\boldsymbol{\alpha}, \boldsymbol{w}_\gamma^*) \\
&= \nabla_{\boldsymbol{\alpha}} \mathcal{H}_\gamma(\boldsymbol{\alpha}, \boldsymbol{w}_\gamma^*) + \nabla_{\boldsymbol{\alpha}} \boldsymbol{w}_\gamma^{*\top} \nabla_{\boldsymbol{w}} \mathcal{H}_\gamma(\boldsymbol{\alpha}, \boldsymbol{w}_\gamma^*) \\
&= \nabla_{\boldsymbol{\alpha}} \mathcal{H}_\gamma(\boldsymbol{\alpha}, \boldsymbol{w}_\gamma^*) \\
&= \gamma \left( \nabla_{\boldsymbol{\alpha}} \mathcal{L}_{\text{train}}(\boldsymbol{\alpha}, \boldsymbol{w}_\gamma^*) - \nabla_{\boldsymbol{\alpha}} \mathcal{L}_{\text{train}}(\boldsymbol{\alpha}, \boldsymbol{w}^*) - \nabla_{\boldsymbol{\alpha}} \boldsymbol{w}_\gamma^{*\top} \nabla_{\boldsymbol{w}} \mathcal{L}_{\text{train}}(\boldsymbol{\alpha}, \boldsymbol{w}^*) \right) \\
&= \gamma \left( \nabla_{\boldsymbol{\alpha}} \mathcal{L}_{\text{train}}(\boldsymbol{\alpha}, \boldsymbol{w}_\gamma^*) - \nabla_{\boldsymbol{\alpha}} \mathcal{L}_{\text{train}}(\boldsymbol{\alpha}, \boldsymbol{w}^*) - \nabla_{\boldsymbol{\alpha w}}^2 \mathcal{L}_{\text{train}}(\boldsymbol{\alpha}, \boldsymbol{w}^*)(\boldsymbol{w}_\gamma^* - \boldsymbol{w}^*) \right) \\
&\quad - \nabla_{\boldsymbol{\alpha}} \boldsymbol{w}^{*\top} \left( \nabla_{\boldsymbol{w}} \mathcal{L}_{\text{val}}(\boldsymbol{w}^*) + \gamma \nabla_{\boldsymbol{w}} \mathcal{L}_{\text{train}}(\boldsymbol{\alpha}, \boldsymbol{w}^*) \right) \\
&\quad + \nabla_{\boldsymbol{\alpha}} \boldsymbol{w}^{*\top} \nabla_{\boldsymbol{w}} \mathcal{L}_{\text{val}}(\boldsymbol{w}^*) + \gamma \nabla_{\boldsymbol{\alpha w}}^2 \mathcal{L}_{\text{train}}(\boldsymbol{\alpha}, \boldsymbol{w}^*)(\boldsymbol{w}_\gamma^* - \boldsymbol{w}^*).
\end{aligned}
\tag{14}
$$

Besides that, we have

$$
\begin{aligned}
&\nabla_{\boldsymbol{\alpha}} \boldsymbol{w}^{*\top} \left( \nabla_{\boldsymbol{w}} \mathcal{L}_{\text{val}}(\boldsymbol{w}^*) + \gamma \nabla_{\boldsymbol{w}} \mathcal{L}_{\text{train}}(\boldsymbol{\alpha}, \boldsymbol{w}^*) \right) \\
&= \nabla_{\boldsymbol{\alpha}} \boldsymbol{w}^{*\top} \left( \nabla_{\boldsymbol{w}} \mathcal{L}_{\text{val}}(\boldsymbol{w}^*) - \nabla_{\boldsymbol{w}} \mathcal{L}_{\text{val}}(\boldsymbol{w}_\gamma^*) \right) \\
&\quad + \gamma \nabla_{\boldsymbol{\alpha}} \boldsymbol{w}^{*\top} \left( \nabla_{\boldsymbol{w}} \mathcal{L}_{\text{train}}(\boldsymbol{\alpha}, \boldsymbol{w}^*) - \nabla_{\boldsymbol{w}} \mathcal{L}_{\text{train}}(\boldsymbol{\alpha}, \boldsymbol{w}_\gamma^*) + \nabla_{\boldsymbol{ww}}^2 \mathcal{L}_{\text{train}}(\boldsymbol{\alpha}, \boldsymbol{w}^*)(\boldsymbol{w}_\gamma^* - \boldsymbol{w}^*) \right) \\
&\quad + \gamma \nabla_{\boldsymbol{\alpha w}}^2 \mathcal{L}_{\text{train}}(\boldsymbol{\alpha}, \boldsymbol{w}^*)(\boldsymbol{w}_\gamma^* - \boldsymbol{w}^*),
\end{aligned}
\tag{15}
$$

due to the optimal conditions $\nabla_{\boldsymbol{w}} \mathcal{L}_{\text{val}}(\boldsymbol{w}_\gamma^*) + \gamma \nabla_{\boldsymbol{w}} \mathcal{L}_{\text{train}}(\boldsymbol{\alpha}, \boldsymbol{w}_\gamma^*) = 0$, and $\nabla_{\boldsymbol{\alpha w}}^2 \mathcal{L}_{\text{train}}(\boldsymbol{\alpha}, \boldsymbol{w}^*) + \nabla_{\boldsymbol{\alpha}} \boldsymbol{w}^{*\top} \nabla_{\boldsymbol{ww}}^2 \mathcal{L}_{\text{train}}(\boldsymbol{\alpha}, \boldsymbol{w}^*) = 0$. Combining the two above equations and (13), we get

$$
\begin{aligned}
&\left\| \frac{\partial}{\partial \boldsymbol{\alpha}} \mathcal{H}_\gamma(\boldsymbol{\alpha}, \boldsymbol{w}_\gamma^*) - \nabla_{\boldsymbol{\alpha}} \mathcal{H}(\boldsymbol{\alpha}) \right\| \\
&\leq \left\| \gamma \left( \nabla_{\boldsymbol{\alpha}} \mathcal{L}_{\text{train}}(\boldsymbol{\alpha}, \boldsymbol{w}_\gamma^*) - \nabla_{\boldsymbol{\alpha}} \mathcal{L}_{\text{train}}(\boldsymbol{\alpha}, \boldsymbol{w}^*) - \nabla_{\boldsymbol{\alpha w}}^2 \mathcal{L}_{\text{train}}(\boldsymbol{\alpha}, \boldsymbol{w}^*)(\boldsymbol{w}_\gamma^* - \boldsymbol{w}^*) \right) \right\| \\
&\quad + \left\| \gamma \nabla_{\boldsymbol{\alpha}} \boldsymbol{w}^{*\top} \left( \nabla_{\boldsymbol{w}} \mathcal{L}_{\text{train}}(\boldsymbol{\alpha}, \boldsymbol{w}^*) - \nabla_{\boldsymbol{w}} \mathcal{L}_{\text{train}}(\boldsymbol{\alpha}, \boldsymbol{w}_\gamma^*) + \nabla_{\boldsymbol{ww}}^2 \mathcal{L}_{\text{train}}(\boldsymbol{\alpha}, \boldsymbol{w}^*)(\boldsymbol{w}_\gamma^* - \boldsymbol{w}^*) \right) \right\| \\
&\quad + \left\| \nabla_{\boldsymbol{\alpha}} \boldsymbol{w}^{*\top} \left( \nabla_{\boldsymbol{w}} \mathcal{L}_{\text{val}}(\boldsymbol{w}^*) - \nabla_{\boldsymbol{w}} \mathcal{L}_{\text{val}}(\boldsymbol{w}_\gamma^*) \right) \right\| \\
&\leq \gamma H \left( \frac{\rho}{\lambda} + 1 \right) \| \boldsymbol{w}^* - \boldsymbol{w}_\gamma^* \|^2 + \frac{\rho L}{\lambda} \| \boldsymbol{w}^* - \boldsymbol{w}_\gamma^* \|,
\end{aligned}
\tag{16}
$$

due to the bounded Hessian Assumption 3, Smoothness Assumption 2, and Lipchitz Assumption 4. Then, due to the PL condition 1 and Smoothness Assumption 2, we know there exists a $\boldsymbol{w}_\gamma^*$ (the projection of $\boldsymbol{w}^*$ to $S_\gamma^*(\boldsymbol{\alpha})$) satisfies

$$
\begin{aligned}
\| \boldsymbol{w}^* - \boldsymbol{w}_\gamma^* \| &\leq \frac{1}{\mu} \| \nabla_{\boldsymbol{w}} \mathcal{L}_{\text{train}}(\boldsymbol{\alpha}, \boldsymbol{w}_\gamma^*) \| \\
&\leq \frac{1}{\gamma \mu} \left( \| \nabla_{\boldsymbol{w}} \mathcal{L}_{\text{val}}(\boldsymbol{w}_\gamma^*) + \gamma \nabla_{\boldsymbol{w}} \mathcal{L}_{\text{train}}(\boldsymbol{w}_\gamma^*) \| + \| \nabla_{\boldsymbol{w}} \mathcal{L}_{\text{val}}(\boldsymbol{w}_\gamma^*) \| \right) \\
&= \frac{\| \nabla_{\boldsymbol{w}} \mathcal{L}_{\text{val}}(\boldsymbol{w}_\gamma^*) \|}{\gamma \mu} \\
&\leq \frac{B}{\gamma \mu}.
\end{aligned}
\tag{17}
$$

Combining this with inequality (16), we obtain the conclusion under such $\boldsymbol{w}_\gamma^*$. Finally, due to the Lemma A.5 in [26], we know that $\nabla_{\boldsymbol{\alpha}} \mathcal{H}_\gamma(\boldsymbol{\alpha}, \boldsymbol{w}_\gamma^*)$ is invariant over $\boldsymbol{w}_\gamma^* \in S_\gamma^*(\boldsymbol{\alpha})$, we prove our conclusion. $\qquad \square$

From Lemma 1, we know that the gap between the gradients of original problem $\mathcal{H}(\boldsymbol{\alpha})$ and its Lagrange version $\mathcal{H}_\gamma(\boldsymbol{\alpha}, \boldsymbol{w}_\gamma^*(\boldsymbol{\alpha}))$ can be extremely small by invoking penalty parameter $\gamma \to \infty$. Thus, it implies that we can compute the gradient of the Lagrange problem to implement the gradient-based method. Next, we illustrate a useful lemma, which characterizes the Lipschitz continuity of $\boldsymbol{w}_\gamma^*(\boldsymbol{\alpha})$ and $\boldsymbol{w}^*(\boldsymbol{\alpha})$ to $\boldsymbol{\alpha}$.

**Lemma 2.** *Under Assumptions 1 and 2, for any $\boldsymbol{\alpha}_1, \boldsymbol{\alpha}_2 \in \mathcal{A}$, it holds*

$$\|\boldsymbol{w}^*(\boldsymbol{\alpha}_1) - \boldsymbol{w}^*(\boldsymbol{\alpha}_2)\| \leq \frac{L}{\mu}\|\boldsymbol{\alpha}_1 - \boldsymbol{\alpha}_2\|, \tag{18}$$

*for any $\boldsymbol{w}^*(\boldsymbol{\alpha}_1) \in S^*(\boldsymbol{\alpha})$ and $\boldsymbol{w}^*(\boldsymbol{\alpha}_2) \in S^*(\boldsymbol{\alpha}_2)$ satisfies $\boldsymbol{w}^*(\boldsymbol{\alpha}_2) = \arg\min_{\boldsymbol{w} \in S^*(\boldsymbol{\alpha}_2)} \|\boldsymbol{w} - \boldsymbol{w}^*(\boldsymbol{\alpha}_1)\|$. On the other hand, it holds*

$$\|\boldsymbol{w}_\gamma^*(\boldsymbol{\alpha}_1) - \boldsymbol{w}_\gamma^*(\boldsymbol{\alpha}_2)\| \leq \frac{\gamma L}{\mu_\gamma}\|\boldsymbol{\alpha}_1 - \boldsymbol{\alpha}_2\| \tag{19}$$

*for any $\boldsymbol{w}_\gamma^*(\boldsymbol{\alpha}_1) \in S_\gamma^*(\boldsymbol{\alpha}_1)$ and $\boldsymbol{w}_\gamma^*(\boldsymbol{\alpha}_2) = \arg\min_{\boldsymbol{w} \in S_\gamma^*(\boldsymbol{\alpha}_1)} \|\boldsymbol{w} - \boldsymbol{w}_\gamma^*(\boldsymbol{\alpha}_1)\|$.*

*Proof.* The two conclusions are directly obtained from Lemma A.3 in [26], we prove the second conclusion as an example. The proof can be similarly generalized to the first conclusion. Due to the formulation of $\nabla_{\boldsymbol{w}}\mathcal{H}_\gamma(\boldsymbol{\alpha}, \boldsymbol{w})$, we know

$$\|\nabla_{\boldsymbol{w}}\mathcal{H}_\gamma(\boldsymbol{\alpha}_1, \boldsymbol{w}_\gamma^*(\boldsymbol{\alpha}_2))\| = \|\nabla_{\boldsymbol{w}}\mathcal{H}_\gamma(\boldsymbol{\alpha}_1, \boldsymbol{w}_\gamma^*(\boldsymbol{\alpha}_2)) - \nabla_{\boldsymbol{w}}\mathcal{H}_\gamma(\boldsymbol{\alpha}_2, \boldsymbol{w}_\gamma^*(\boldsymbol{\alpha}_2))\|$$
$$= \gamma \left\|\nabla_{\boldsymbol{w}}\mathcal{L}_{\text{train}}(\boldsymbol{\alpha}_1, \boldsymbol{w}_\gamma^*(\boldsymbol{\alpha}_2)) - \nabla_{\boldsymbol{w}}\mathcal{L}_{\text{train}}(\boldsymbol{\alpha}_2, \boldsymbol{w}_\gamma^*(\boldsymbol{\alpha}_2))\right\| \tag{20}$$
$$\leq \gamma L\|\boldsymbol{\alpha}_1 - \boldsymbol{\alpha}_2\|,$$

by Assumption 2. On the other hand, by invoking the Assumption 1 and (17), we get

$$\|\nabla_{\boldsymbol{w}}\mathcal{H}_\gamma(\boldsymbol{\alpha}_1, \boldsymbol{w}_\gamma^*(\boldsymbol{\alpha}_2))\| = \left\|\nabla_{\boldsymbol{w}}\mathcal{L}_{\text{val}}(\boldsymbol{w}_\gamma^*(\boldsymbol{\alpha}_2)) + \gamma\nabla_{\boldsymbol{w}}\mathcal{L}_{\text{train}}(\boldsymbol{\alpha}_1, \boldsymbol{w}_\gamma^*(\boldsymbol{\alpha}_2))\right\|$$
$$\geq \mu_\gamma\|\boldsymbol{w}_\gamma^*(\boldsymbol{\alpha}_2) - \boldsymbol{w}_\gamma^*(\boldsymbol{\alpha}_1)\|, \tag{21}$$

where $\boldsymbol{w}_\gamma^*(\boldsymbol{\alpha}_1)$ is the projection of $\boldsymbol{w}^*(\boldsymbol{\alpha}_2)$ to $S_\gamma^*(\boldsymbol{\alpha}_1)$. By combining the two above inequalities, we obtain our second conclusion. $\square$

From this lemma, we can obtain the following Lipschitz smoothness of $\nabla_{\boldsymbol{\alpha}}\mathcal{H}_\gamma(\boldsymbol{\alpha}, \boldsymbol{w}_\gamma^*(\boldsymbol{\alpha}))$ w.r.t. $\boldsymbol{\alpha}$ by the following Lemma. It worth noting that $\nabla_{\boldsymbol{\alpha}}\mathcal{H}_\gamma(\boldsymbol{\alpha}, \boldsymbol{w}_\gamma^*(\boldsymbol{\alpha}))$ is invariant over $\boldsymbol{w}_\gamma^*(\boldsymbol{\alpha}) \in S_\gamma^*(\boldsymbol{\alpha})$ as discussed in the proof of Lemma 1. Thus, the $\nabla_{\boldsymbol{\alpha}}\mathcal{H}_\gamma(\boldsymbol{\alpha}, \boldsymbol{w}_\gamma^*(\boldsymbol{\alpha}))$ is well-defined.

**Lemma 3.** *Under Assumptions 1 and 2, $\nabla_{\boldsymbol{\alpha}}\mathcal{H}_\gamma(\boldsymbol{\alpha}, \boldsymbol{w}_\gamma^*(\boldsymbol{\alpha}))$ has semi-Lipschitz gradient such that*

$$\|\nabla_{\boldsymbol{\alpha}}\mathcal{H}_\gamma(\boldsymbol{\alpha}_1, \boldsymbol{w}_\gamma^*(\boldsymbol{\alpha}_1)) - \nabla_{\boldsymbol{\alpha}}\mathcal{H}_\gamma(\boldsymbol{\alpha}_2, \boldsymbol{w}_\gamma^*(\boldsymbol{\alpha}_2))\| \leq \underbrace{\gamma B\left(\frac{\gamma L}{\mu_\gamma} + \frac{L}{\mu}\right)}_{L_\gamma}\|\boldsymbol{\alpha}_1 - \boldsymbol{\alpha}_2\|. \tag{22}$$

*Proof.* From (14), we see

$$\left\|\nabla_{\boldsymbol{\alpha}}\mathcal{H}_\gamma(\boldsymbol{\alpha}_1, \boldsymbol{w}_\gamma^*(\boldsymbol{\alpha}_1)) - \nabla_{\boldsymbol{\alpha}}\mathcal{H}_\gamma(\boldsymbol{\alpha}_2, \boldsymbol{w}_\gamma^*(\boldsymbol{\alpha}_2))\right\|$$
$$\leq \gamma \left\|\nabla_{\boldsymbol{\alpha}}\mathcal{L}_{\text{train}}(\boldsymbol{\alpha}_1, \boldsymbol{w}_\gamma^*(\boldsymbol{\alpha}_1)) - \nabla_{\boldsymbol{\alpha}}\mathcal{L}_{\text{train}}(\boldsymbol{\alpha}_2, \boldsymbol{w}_\gamma^*(\boldsymbol{\alpha}_2))\right\|$$
$$+ \gamma \left\|\nabla_{\boldsymbol{\alpha}}\mathcal{L}_{\text{train}}(\boldsymbol{\alpha}_2, \boldsymbol{w}^*(\boldsymbol{\alpha}_2)) - \nabla_{\boldsymbol{\alpha}}\mathcal{L}_{\text{train}}(\boldsymbol{\alpha}_1, \boldsymbol{w}^*(\boldsymbol{\alpha}_1))\right\| \tag{23}$$
$$\leq \gamma B \left(\|\boldsymbol{w}_\gamma(\boldsymbol{\alpha}_1) - \boldsymbol{w}_\gamma(\boldsymbol{\alpha}_2)\| + \|\boldsymbol{w}(\boldsymbol{\alpha}_1) - \boldsymbol{w}(\boldsymbol{\alpha}_2)\|\right)$$
$$\leq \gamma B \left(\frac{\gamma L}{\mu_\gamma} + \frac{L}{\mu}\right)\|\boldsymbol{\alpha}_1 - \boldsymbol{\alpha}_2\|,$$

which proves our conclusion. $\square$

Notably, for the original optimization problem, there exists a constraint $\boldsymbol{\alpha} \in \mathcal{A}$. Thus, to prove the first order stationary condition of constrain problem $\min_{\boldsymbol{\alpha} \in \mathcal{A}} \mathcal{H}_{\boldsymbol{\alpha}}$, we consider the generalized projected gradient stable condition, i.e., $\frac{1}{\eta_{\boldsymbol{\alpha}}}\|\boldsymbol{\alpha}^{(t)} - \Pi_{\mathcal{A}}(\boldsymbol{\alpha}^{(t)} - \eta_{\boldsymbol{\alpha}}\nabla_{\boldsymbol{\alpha}}\mathcal{H}(\boldsymbol{\alpha}^{(t)}))\| \leq \epsilon$ for some small positive $\epsilon$. This is a standard first-order stationary condition for Non-convex optimization problem with constrains [30, 42, 9]. Due to Lemma 1, we know that

$$\frac{1}{\eta_{\boldsymbol{\alpha}}}\left\|\boldsymbol{\alpha}^{(t)} - \Pi_{\mathcal{A}}(\boldsymbol{\alpha}^{(t)} - \eta_{\boldsymbol{\alpha}}\nabla_{\boldsymbol{\alpha}}\mathcal{H}(\boldsymbol{\alpha}^{(t)}))\right\| - \frac{1}{\eta_{\boldsymbol{\alpha}}}\left\|\boldsymbol{\alpha}^{(t)} - \Pi_{\mathcal{A}}(\boldsymbol{\alpha}^{(t)} - \eta_{\boldsymbol{\alpha}}\nabla_{\boldsymbol{\alpha}}\mathcal{H}_\gamma(\boldsymbol{\alpha}^{(t)}, \boldsymbol{w}_\gamma^*(\boldsymbol{\alpha}^{(t)})))\right\|$$
$$\leq \frac{1}{\eta_{\boldsymbol{\alpha}}}\left\|\Pi_{\mathcal{A}}(\boldsymbol{\alpha}^{(t)} - \eta_{\boldsymbol{\alpha}}\nabla_{\boldsymbol{\alpha}}\mathcal{H}(\boldsymbol{\alpha}^{(t)})) - \Pi_{\mathcal{A}}(\boldsymbol{\alpha}^{(t)} - \eta_{\boldsymbol{\alpha}}\nabla_{\boldsymbol{\alpha}}\mathcal{H}_\gamma(\boldsymbol{\alpha}^{(t)}, \boldsymbol{w}_\gamma^*(\boldsymbol{\alpha}^{(t)})))\right\|$$
$$\leq \left\|\nabla_{\boldsymbol{\alpha}}\mathcal{H}(\boldsymbol{\alpha}^{(t)}) - \nabla_{\boldsymbol{\alpha}}\mathcal{H}_\gamma(\boldsymbol{\alpha}^{(t)}, \boldsymbol{w}_\gamma^*(\boldsymbol{\alpha}^{(t)}))\right\| \tag{24}$$
$$\leq \mathcal{O}\left(\frac{1}{\gamma}\right),$$

when $\gamma \to \infty$. Here the second inequality is from the Lipschitz continuity of projection operation [9]. Thus, the above inequality indicates that to prove the first-order stationary condition of $\mathcal{H}(\boldsymbol{\alpha})$, it is sufficient to prove the first-order stationary condition of $\mathcal{H}_\gamma(\boldsymbol{\alpha}, \boldsymbol{w})$. Next we present our main theorem, i.e., the formal version of Theorem 1, which shows the convergence result of $\mathcal{H}_{\boldsymbol{\alpha}}(\boldsymbol{\alpha})$ under Algorithm 1.

**Theorem 1.** *For sufficiently large T, under Assumptions 1-5, for the $\boldsymbol{\alpha}^{(t)}$ obtained in Algorithm 1, it holds*

$$\min_{1 \le t \le T} \frac{1}{\eta_{\boldsymbol{\alpha}}} \|\boldsymbol{\alpha}^{(t)} - \Pi_{\mathcal{A}}(\boldsymbol{\alpha}^{(t)} - \eta_{\boldsymbol{\alpha}} \nabla_{\boldsymbol{\alpha}} \mathcal{H}(\boldsymbol{\alpha}^{(t)}))\| \le \mathcal{O}\left(T^{-\frac{1}{4}}\right) \tag{25}$$

*by selecting* $\gamma = T^{\frac{1}{4}}$, $K \ge \dfrac{\log \frac{\mu T^{-1}}{2D(1+\gamma)}}{\log\left(1 - \frac{\mu\gamma}{L_\gamma}\right)}$, $E \ge \min\left\{1, \dfrac{\log \frac{\mu_\gamma T^{-1}}{2D}}{\log\left(1 - \frac{\mu}{L}\right)} - K\right\}$, $\eta_{\boldsymbol{\alpha}} = \frac{1}{4L_\gamma}$, $\eta_{\boldsymbol{w}} = \frac{1}{L_\gamma}$, $\eta_{\boldsymbol{u}} = \frac{1}{L}$.

*Proof.* As mentioned in (24), it is sufficient to prove the first-order stationary condition for $\mathcal{H}_\gamma(\boldsymbol{\alpha}, \boldsymbol{w})$. Due to the Lipchitz smoothness of it, we have

$$\begin{aligned}
&\mathcal{H}_\gamma(\boldsymbol{\alpha}^{(t+1)}, \boldsymbol{w}_\gamma^*(\boldsymbol{\alpha}^{(t+1)})) - \mathcal{H}_\gamma(\boldsymbol{\alpha}^{(t)}, \boldsymbol{w}_\gamma^*(\boldsymbol{\alpha}^{(t)})) \\
&\le \left\langle \nabla_{\boldsymbol{\alpha}} \mathcal{H}_\gamma(\boldsymbol{\alpha}^{(t)}, \boldsymbol{w}_\gamma^*(\boldsymbol{\alpha}^{(t)})), \boldsymbol{\alpha}^{(t+1)} - \boldsymbol{\alpha}^{(t)} \right\rangle + \frac{L_\gamma}{2} \|\boldsymbol{\alpha}^{(t+1)} - \boldsymbol{\alpha}^{(t)}\|^2 \\
&= \left\langle \nabla_{\boldsymbol{\alpha}} \mathcal{F}_\gamma(\boldsymbol{\alpha}^{(t)}, \boldsymbol{w}_K^{(t)}, \boldsymbol{u}_K^{(t)}), \boldsymbol{\alpha}^{(t+1)} - \boldsymbol{\alpha}^{(t)} \right\rangle \\
&\quad + \left\langle \nabla_{\boldsymbol{\alpha}} \mathcal{H}_\gamma(\boldsymbol{\alpha}^{(t)}, \boldsymbol{w}_\gamma^*(\boldsymbol{\alpha}^{(t)})) - \nabla_{\boldsymbol{\alpha}} \mathcal{F}_\gamma(\boldsymbol{\alpha}^{(t)}, \boldsymbol{w}_K^{(t)}, \boldsymbol{u}_K^{(t)}), \boldsymbol{\alpha}^{(t+1)} - \boldsymbol{\alpha}^{(t)} \right\rangle \\
&\quad + \frac{L_\gamma}{2} \left\| \boldsymbol{\alpha}^{(t+1)} - \boldsymbol{\alpha}^{(t)} \right\|^2 \\
&\le \left( \frac{L_\gamma}{2} - \frac{1}{2\eta_{\boldsymbol{\alpha}}} \right) \left\| \boldsymbol{\alpha}^{(t+1)} - \boldsymbol{\alpha}^{(t)} \right\|^2 + \frac{\eta_{\boldsymbol{\alpha}}}{2} \left\| \nabla_{\boldsymbol{\alpha}} \mathcal{H}_\gamma(\boldsymbol{\alpha}^{(t)}, \boldsymbol{w}_\gamma^*(\boldsymbol{\alpha}^{(t)})) - \nabla_{\boldsymbol{\alpha}} \mathcal{F}_\gamma(\boldsymbol{\alpha}^{(t)}, \boldsymbol{w}_K^{(t)}, \boldsymbol{u}_K^{(t)}) \right\|^2,
\end{aligned} \tag{26}$$

where the last inequality is due to the property of projection operator and Jensen's inequality. Let us define $\bar{\boldsymbol{\alpha}}^{(t+1)} = \Pi_{\mathcal{A}}(\boldsymbol{\alpha}^{(t)} - \eta_{\boldsymbol{\alpha}} \nabla_{\boldsymbol{\alpha}} \mathcal{H}_\gamma(\boldsymbol{\alpha}^{(t)}, \boldsymbol{w}_\gamma^*(\boldsymbol{\alpha}^{(t)})))$, we proceed to upper bound the gap between $\|\bar{\boldsymbol{\alpha}}^{(t+1)} - \boldsymbol{\alpha}^{(t)}\|$ and $\|\boldsymbol{\alpha}^{(t+1)} - \boldsymbol{\alpha}^{(t)}\|$. By the triangle inequality

$$\begin{aligned}
&\left| \|\bar{\boldsymbol{\alpha}}^{(t+1)} - \boldsymbol{\alpha}^{(t)}\| - \|\boldsymbol{\alpha}^{(t+1)} - \boldsymbol{\alpha}^{(t)}\| \right| \\
&\le \|\bar{\boldsymbol{\alpha}}^{(t+1)} - \boldsymbol{\alpha}^{(t+1)}\| \\
&\le \left\| \Pi_{\mathcal{A}}\left( \boldsymbol{\alpha}^{(t)} - \eta_{\boldsymbol{\alpha}} \nabla_{\boldsymbol{\alpha}} \mathcal{F}_\gamma(\boldsymbol{\alpha}^{(t)}, \boldsymbol{w}_K^{(t)}, \boldsymbol{u}_K^{(t)}) \right) - \Pi_{\mathcal{A}}\left( \boldsymbol{\alpha}^{(t)} - \eta_{\boldsymbol{\alpha}} \nabla_{\boldsymbol{\alpha}} \mathcal{H}_\gamma(\boldsymbol{\alpha}^{(t)}, \boldsymbol{w}_\gamma^*(\boldsymbol{\alpha}^{(t)})) \right) \right\| \\
&\le \eta_{\boldsymbol{\alpha}} \left\| \nabla_{\boldsymbol{\alpha}} \mathcal{F}_\gamma(\boldsymbol{\alpha}^{(t)}, \boldsymbol{w}_K^{(t)}, \boldsymbol{u}_K^{(t)}) - \nabla_{\boldsymbol{\alpha}} \mathcal{H}_\gamma(\boldsymbol{\alpha}^{(t)}, \boldsymbol{w}_\gamma^*(\boldsymbol{\alpha}^{(t)})) \right\| \\
&\le \eta_{\boldsymbol{\alpha}} \gamma B \|\boldsymbol{w}_K^{(t)} - \boldsymbol{w}_\gamma^*(\boldsymbol{\alpha}^{(t)})\| + \eta_{\boldsymbol{\alpha}} \gamma B \|\boldsymbol{u}_K^{(t)} - \boldsymbol{w}^*(\boldsymbol{\alpha}^{(t)})\|,
\end{aligned} \tag{27}$$

where the last inequality is from the smoothness Assumption 2, $\boldsymbol{w}_\gamma^*(\boldsymbol{\alpha}^{(t)})$ and $\boldsymbol{w}^*(\boldsymbol{\alpha}^{(t)})$ are respectively the projection of $\boldsymbol{w}_K^{(t)}$ and $\boldsymbol{u}_K^{(t)}$ to $S_\gamma^*(\boldsymbol{\alpha}^{(t)})$ and $S^*(\boldsymbol{\alpha}^{(t)})$. Firstly, due to the PL-condition and Lipschitz smoothness of $\mathcal{L}_{\text{train}}$, we have

$$\begin{aligned}
\mathcal{L}_{\text{train}}(\boldsymbol{\alpha}^{(t)}, \boldsymbol{u}_{k+1}^{(t)}) - \mathcal{L}_{\text{train}}(\boldsymbol{\alpha}^{(t)}, \boldsymbol{u}_k^{(t)}) &\le \left\langle \nabla_{\boldsymbol{w}} \mathcal{L}_{\text{train}}(\boldsymbol{\alpha}^{(t)}, \boldsymbol{u}_k^{(t)}), \boldsymbol{u}_{k+1}^{(t)} - \boldsymbol{u}_k^{(t)} \right\rangle + \frac{L}{2} \|\boldsymbol{u}_{k+1}^{(t)} - \boldsymbol{u}_k^{(t)}\|^2 \\
&= -\frac{1}{2L} \|\nabla_{\boldsymbol{w}} \mathcal{L}_{\text{train}}(\boldsymbol{\alpha}^{(t)}, \boldsymbol{u}_k^{(t)})\|^2 \\
&\le -\frac{\mu}{L} \left( \mathcal{L}_{\text{train}}(\boldsymbol{\alpha}^{(t)}, \boldsymbol{u}_k^{(t)}) - \mathcal{L}_{\text{train}}(\boldsymbol{\alpha}^{(t)}, \boldsymbol{w}^*(\boldsymbol{\alpha}^{(t)})) \right),
\end{aligned} \tag{28}$$

which implies

$$\begin{aligned}
\|\boldsymbol{u}_K^{(t)} - \boldsymbol{w}^*(\boldsymbol{\alpha}^{(t)})\|^2 &\le \frac{2}{\mu} \left( 1 - \frac{\mu}{L} \right)^{K+E} \left( \mathcal{L}_{\text{train}}(\boldsymbol{\alpha}^{(t)}, \boldsymbol{u}_0^{(t)}) - \mathcal{L}_{\text{train}}(\boldsymbol{\alpha}^{(t)}, \boldsymbol{w}^*(\boldsymbol{\alpha}^{(t)})) \right) \\
&\le \frac{2}{\mu} \left( 1 - \frac{\mu}{L} \right)^{K+E} D \\
&= \mathcal{O}\left( \frac{1}{T} \right).
\end{aligned} \tag{29}$$

due to the selection of $K$ and $E$ and the PL condition of $\mathcal{L}_{\text{train}}(\boldsymbol{\alpha}, \boldsymbol{w})$ (which implies the error bound condition [17]). Similarly, we can prove that

$$
\begin{aligned}
\|\boldsymbol{w}_K^{(t)} - \boldsymbol{w}_\gamma^*(\boldsymbol{\alpha}^{(t)})\|^2 &\leq \frac{2}{\mu_\gamma}\left(1 - \frac{\mu}{L}\right)^K \left(\mathcal{H}_\gamma(\boldsymbol{\alpha}^{(t)}, \boldsymbol{w}_K^{(t)}) - \mathcal{H}_\gamma(\boldsymbol{\alpha}^{(t)}, \boldsymbol{w}_\gamma^*(\boldsymbol{\alpha}^{(t)}))\right) \\
&\leq \frac{2}{\mu_\gamma}\left(1 - \frac{\mu_\gamma}{L_\gamma}\right)^K (1 + \gamma) D \\
&= \mathcal{O}\left(\frac{1}{T}\right).
\end{aligned} \tag{30}
$$

Combining (26) (27), (29), and (30), we have

$$
\mathcal{H}_\gamma(\boldsymbol{\alpha}^{(t+1)}, \boldsymbol{w}_\gamma^*(\boldsymbol{\alpha}^{(t+1)})) - \mathcal{H}_\gamma(\boldsymbol{\alpha}^{(t)}, \boldsymbol{w}_\gamma^*(\boldsymbol{\alpha}^{(t)})) \leq -\frac{1}{4\eta_{\boldsymbol{\alpha}}}\left\|\boldsymbol{\alpha}^{(t+1)} - \boldsymbol{\alpha}^{(t)}\right\|^2 + \mathcal{O}\left(\frac{\eta_{\boldsymbol{\alpha}}\gamma^2}{T^{\frac{4}{3}}}\right), \tag{31}
$$

so that, by combining (27), we get

$$
\begin{aligned}
\frac{1}{T}\sum_{t=1}^{T}\frac{1}{\eta_{\boldsymbol{\alpha}}^2}\left\|\bar{\boldsymbol{\alpha}}^{(t+1)} - \boldsymbol{\alpha}^{(t)}\right\|^2 &\leq \frac{1}{T}\sum_{t=1}^{T}\frac{2}{\eta_{\boldsymbol{\alpha}}^2}\left[\left\|\boldsymbol{\alpha}^{(t+1)} - \boldsymbol{\alpha}^{(t)}\right\|^2 + \left|\|\bar{\boldsymbol{\alpha}}^{(t+1)} - \boldsymbol{\alpha}^{(t)}\| - \|\boldsymbol{\alpha}^{(t+1)} - \boldsymbol{\alpha}^{(t)}\|\right|^2\right] \\
&\leq \frac{\mathcal{H}_\gamma(\boldsymbol{\alpha}^{(0)}, \boldsymbol{w}_\gamma^{(*)}(\boldsymbol{\alpha}^{(0)})) - \inf_{\boldsymbol{\alpha}}\mathcal{H}_\gamma(\boldsymbol{\alpha}^{(0)}, \boldsymbol{w}_\gamma^{(*)}(\boldsymbol{\alpha}^{(0)}))}{\eta_{\boldsymbol{\alpha}} T} + \mathcal{O}\left(\frac{\gamma^2}{T}\right) \\
&= \mathcal{O}\left(\frac{\gamma^2}{T}\right) \\
&= \mathcal{O}\left(T^{-\frac{1}{2}}\right).
\end{aligned}
$$
$$\tag{32}$$

Due to the definition of $\bar{\boldsymbol{\alpha}}^{(t+1)}$, combining this with (24), and the value of $\gamma$ proves our conclusion. $\qquad\square$

## B    Proof of the Proposition

**Proposition 1.** *Assume $\mathcal{L}_{train}^m = \mathcal{L}_{val}^m$, the uniform mixture ratio $\bar{\alpha}_m = \frac{1}{M}$ for $m = 1, \ldots, M$ constitutes a valid solution of* (1).

*Proof.* Let $\mathcal{L}_{\text{train}}^m = \mathcal{L}_{\text{val}}^m = \mathbb{E}_{\text{data}}[\mathcal{L}_{\text{data}}^m] := \mathcal{L}^m$. To establish the above, it suffices to show:

$$
\sum_{m=1}^{M}\mathcal{L}_{\text{val}}^m(\boldsymbol{w}^*(\bar{\boldsymbol{\alpha}})) \leq \sum_{m=1}^{M}\mathcal{L}_{\text{val}}^m(\boldsymbol{w}^*(\boldsymbol{\alpha}))
$$

which is equivalent to:

$$
\sum_{m=1}^{M}\bar{\alpha}_m\mathcal{L}^m(\boldsymbol{w}^*(\bar{\boldsymbol{\alpha}})) \leq \sum_{m=1}^{M}\bar{\alpha}_m\mathcal{L}^m(\boldsymbol{w}^*(\boldsymbol{\alpha})) \tag{33}
$$

Because $\boldsymbol{w}^*(\bar{\boldsymbol{\alpha}})$ is the minimizer of the inner-level problem under uniform weighting, we have:

$$
\sum_{m=1}^{M}\bar{\alpha}_m\mathcal{L}^m(\boldsymbol{w}^*(\bar{\boldsymbol{\alpha}})) \leq \sum_{m=1}^{M}\bar{\alpha}_m\mathcal{L}^m(\boldsymbol{w}) \tag{34}
$$

for any $\boldsymbol{w}$. In particular, by choosing $\boldsymbol{w} = \boldsymbol{w}^*(\boldsymbol{\alpha})$, the right-hand side becomes the expression in (33), completing the proof. $\qquad\square$

## C    Implementations and Hyper-parameters

**Implementation Details**    We use the gradient descent optimizer for the $K$ step $\boldsymbol{u}$ and $\boldsymbol{w}$ update. Their learning rates $\eta_{\boldsymbol{u}}^K$ and $\eta_{\boldsymbol{w}}^K$ are kept the same so that the domain-wise loss difference $\mathcal{L}_{\text{train}}^m(\boldsymbol{w}_K) - \mathcal{L}_{\text{train}}^m(\boldsymbol{u}_K)$ faithfully reflects the gain of incorporating the additional validation data. For the $E$ step $\boldsymbol{u}$ update, we utilize the Adam optimizer for fast training.

Table 6: Hyper-parameters of TANDEM for different application scenarios

| | Data Aundent | Data Restricted | | | Supervised Fine-tuning |
|---|---|---|---|---|---|
| | GPT-like 160M | 160M | 410M | 1B | Qwen2-0.5B |
| Batch Size | 8 | 128 | 128 | 128 | 32 |
| Learning Rate $\eta_u^E$ | 5e-5 | 5e-4 | 5e-4 | 5e-4 | 4e-6 |
| Learning Rate $\eta_u^K$ | 1e-2 | 1e-2 | 1e-2 | 1e-2 | 1e-2 |
| Learning Rate $\eta_w^K$ | 1e-2 | 1e-2 | 1e-2 | 1e-2 | 1e-2 |
| Learning Rate $\eta_\alpha$ | 2e-3 | 4e-3 | 4e-3 | 4e-3 | 4e-3 |
| Learning Rate Scheduler | Cosine | Cosine | Cosine | Cosine | Cosine |
| Penalty $\gamma$ | 1.0 | 1.0 | 1.0 | 1.0 | 1.0 |
| Probing Steps K | 5 | 5 | 5 | 5 | 10 |
| Free Training Steps E | 20 | 5 | 5 | 5 | 10 |
| Total Steps (w.r.t $u$) | 40000 | 5000 | 5000 | 5000 | 5000 |
| Context Length | 2048 | 512 | 512 | 512 | 512 |
| Weight Decay | 1e-2 | 1e-2 | 1e-2 | 1e-2 | 1e-2 |
| Gradient Clipping | 1.0 | 1.0 | 1.0 | 1.0 | 1.0 |

Table 7: Statistics of the SFT data.

| Method | Num. Sample |
|---|---|
| Textual Entailment | 79332 |
| Answer Verification | 13195 |
| Text Matching | 47297 |
| Information Extraction | 31053 |
| Word Extraction | 17294 |
| Text Categorization | 89572 |

Table 8: Statistics of the task-level SFT data.

| Method | Num. Sample |
|---|---|
| SQuAD1.1 | 6498 |
| AMRSum | 6500 |
| MuTual | 6500 |
| SemEval | 5996 |
| SST2 | 6495 |
| BoolQ | 6500 |

**Hyper-parameter settings**  The detailed hyper-parameters for the TANDEM algorithms in the conventional data-abundant pretraining scenario, the data-restricted training scenario, and the supervised fine-tuning are shown in Table 6.

## D  Supervised Learning Data Statistics

In the supervised fine-tuning case, we use 6 major categories from Natural Instructions [25, 35]: Textual Entailment, Answer Verification, Text Matching, Information Extraction, Word Extraction, and Text Categorization, each constitutes a task cluster. This corpus comprises 99 tasks ranging from open-ended text generation, multiple choice, and True/False tasks. The statistics are given in Table 7.

Besides, we delve into a more fine-grained task-level SFT case, and select 6 tasks SQuAD1.1, AMRSum, MuTual, SemEval, SST2, and BoolQ. The statistics are given in Table 8. For the text generation tasks SQuAD1.1 and AMRSum, we report the Rouge-L [22] score. For the multi-choice tasks (MuTual, SemEval) and Yes/No tasks (SST2, BoolQ), we focus on the accuracy. The test loss is reported as well. We experiment with Qwen2-500M with $K = 20$, $E = 10$, batch size 32, and context length 512 for 2000 steps. The result is shown in Table 9. In this case, the improvement in test loss is more significant than that in the final evaluation metrics, likely due to the imperfect alignment between them.

Table 9: Comparison for the 500M Qwen-2 model on a mixture of 6 tesk-level SFT tasks.

| Method | SQuAD1.1 | AMRSum | MuTual | SemEval | SST2 | BoolQ | Test Loss ↓ | Avg. Metric ↑ |
|---|---|---|---|---|---|---|---|---|
| Uniform | 72.62 | 45.18 | 72.72 | **89.75** | 87.75 | 80.29 | 0.591 | 74.72 |
| DoReMi | 71.40 | 43.40 | 70.97 | 88.50 | 87.00 | 77.43 | 0.686 | 73.11 |
| DoGE | 71.26 | 44.81 | 71.67 | 89.00 | 88.30 | **81.04** | 0.563 | 74.35 |
| Skill-It | 72.14 | 44.21 | 73.07 | 89.40 | 88.40 | 80.34 | 0.539 | 74.60 |
| Aioli | 72.35 | 45.01 | 73.57 | 89.10 | 88.10 | 79.64 | 0.542 | 74.63 |
| TANDEM | **72.73** | **45.19** | **73.77** | 89.70 | **88.70** | 80.04 | **0.508** | **75.03** |

Table 10: Comparison of models of different sizes in the data-abundant pretraining scenario and data restricted scenario with standard deviation.

| | Data Aundant Regime | Data Restricted Regime | | |
| | 160M Avg. | 160M Avg. | 410M Avg. | 1B Avg. |
|---|---|---|---|---|
| Uniform | $25.74_{\pm0.13}$ | $31.53_{\pm0.11}$ | $29.59_{\pm0.02}$ | $29.91_{\pm0.09}$ |
| DoReMi | $28.32_{\pm0.12}$ | $36.91_{\pm0.09}$ | $54.61_{\pm0.60}$ | $56.53_{\pm0.53}$ |
| DoGE | $26.60_{\pm0.21}$ | $30.10_{\pm0.05}$ | $27.45_{\pm0.02}$ | - |
| Skill-It | $25.87_{\pm0.07}$ | $29.24_{\pm0.08}$ | $27.70_{\pm0.03}$ | $27.15_{\pm0.17}$ |
| Aioli | $25.66_{\pm0.14}$ | $31.19_{\pm0.25}$ | $28.79_{\pm0.06}$ | $28.07_{\pm0.03}$ |
| TANDEM | $\mathbf{25.43}_{\pm0.15}$ | $\mathbf{28.07}_{\pm0.07}$ | $\mathbf{25.00}_{\pm0.01}$ | $\mathbf{24.35}_{\pm0.03}$ |

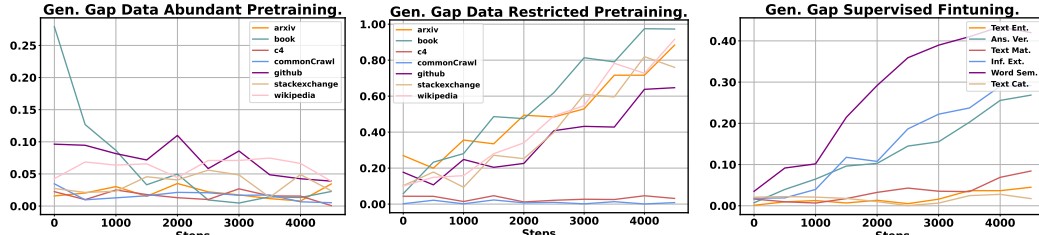

Figure 9: Step-wise generalization gap $|\mathcal{L}_{\text{val}}^m - \mathcal{L}_{\text{train}}^m|$ evolution under three scenarios. (a) data-abundant pretraining (b) data-restricted pretraining and (c) supervised fine-tuning.

# E  Comparison with Standard Deviation

We test each method in Table 3, Table 4 3 times. The averaged perplexity and standard deviation are reported in Table 10.

# F  Additional Experiments

## F.1  Generalization Gap Analyses

To validate our analysis in Section 2.4, we plot the change of generalization gap $|\mathcal{L}_{\text{val}}^m - \mathcal{L}_{\text{train}}^m|$ during training under the three circumstances Figure 9. In the data-abundant scenario, the generalization gaps of each domain do not change much and are kept close to 0. According to the analysis in Section 2.4, Uniform is a valid solution in this scenario. In the data-restricted scenario, we see an increase in the generalization gap, particularly on small domains (Arxiv, Book, Wikipedia), because the sample repetition kicks in. In the large domains (C4, CommonCrawl) where there is no data repetition, the generalization remains small. SFT is of the same case. Generalization gap increase in small tasks (Word Semantics, Information Extraction), while kept small for large tasks (Text Categorization, Text Matching, Textual Entailment). The results on all three scenarios validate our analysis in Section 2.4.

## F.2  Larger Scale Experiments

To validate the effectiveness of TANDEM in practice, we conduct experiments to test its performance under more realistic conditions. This includes scaling up both the dataset size and the model size.

Table 11: Comparison for larger data scale (6B tokens).

| | Arxiv | Book | C4 | CC | Github | Stackexchange | Wikipedia | Average |
|---|---|---|---|---|---|---|---|---|
| Uniform | 10.28 | 32.37 | 38.53 | 34.57 | 5.86 | 10.76 | 15.26 | 17.09 |
| TANDEM | 10.26 | 29.59 | 32.52 | 29.52 | 6.01 | 10.68 | 16.02 | 16.25 |

Table 12: Comparison for larger (3B) model in the data restricted regime.

|  | Arxiv | Book | C4 | CC | Github | Stackexchange | Wikipedia | Average |
|---|---|---|---|---|---|---|---|---|
| Uniform | 17.20 | 53.83 | 60.14 | 55.45 | 9.85 | 22.08 | 41.76 | 31.03 |
| TANDEM | 15.73 | 42.85 | 43.60 | 39.59 | 7.49 | 17.26 | 29.20 | 23.81 |

Table 13: Comparison for the larger (3B) model in the supervised fine-tuning.

| Method | Textual Ent. | Answer Ver. | Text Mat. | Inf. Ext. | Word Sem. | Text Cat. | Avg. Metric ↑ | Test Loss ↓ |
|---|---|---|---|---|---|---|---|---|
| Uniform | 91.9 | 74.8 | 88.8 | 79.2 | 88.8 | 85.5 | 84.9 | 0.189 |
| TANDEM | 92.4 | 76.9 | 89.6 | 80.1 | 88.0 | 87.9 | 85.8 | 0.174 |

**Larger Data Scale** We train the 160M models on the full 6B version of sampled SlimPajama, which constitutes a more practical past Chinchilla [14] (Chinchilla optimal requires $\approx$3.2B data for the 160M model) case. From Table 11, we see that TANDEM still outperforms Uniform, though it seems that the improvement ($\sim 1$) in this case is not as significant as in the 300M data case ($\sim 3.5$). As the training proceeds and the perplexity goes down, further gains become inherently harder to achieve. This result is still significant, showing that careful mixing still matters in this "over-trained" case.

**Larger Model Scale** We conduct DMO with 3B models in the data-restricted scenario (Pythia 2.9B) and SFT (Qwen-2.5-3B). Further scaling up necessitates engineering upgrades to fit the two models ($u$ and $w$) within the 80GB memory, while also demanding substantial time and computational resources. We skip the data-abundant scenario where Uniform is already a valid solution. From Table 12, we see that as the model goes larger, inappropriate mixture ratios (Uniform) lead to more obvious negative outcomes(e.g. severe overfitting on the overly sampled small domains). While TANDEM consistently generates proper mixture ratios. For SFT (Table 13), TANDEM still outperforms the Uniform baseline, showing its effectiveness.

### F.3 Sensitivity of $K$ and $E$

$K$: $K$ is the number of probing steps used to estimate the proper update direction ((3) $\sim$ (5)). We conduct a sensitivity analysis with $K = 1, 3, 5, 10$ to see how it will affect the final model performance. From Table 14, we see that too small $K$ may results in less fidel $\alpha$ update direction, thus sub-optimal data mixture ratio and higher perplexity. When $K$ is large enough for $\alpha$ update probing, increasing $K$ will not induce further benefits. In our experiments, for relatively stable pretraining, we use a smaller $K = 5$ in the data-abundant and data-restricted scenarios. The gradient variance in SFT is much higher, so we use larger $K = 10$.

$E$: Given the total number of training steps (amounts of data to be used), $E$ determines the number of updates during DMO: $T = \frac{total\ steps}{E}$. We test $E = 1, 5, 10$ in the 160M data-restricted scenario. From Table 15 we see that TANDEM is not sensitive to $E$ as long as $T$ is large enough so that $\alpha$ is sufficiently updated. In our experiments, for the data-abundant scenario, we chose $E = 20$ so that the mixture ratio is updated for $T = 2000$ steps. As one $\alpha$ update requires additional $K$ steps $w$ updates, a larger $T$ means higher additional computational cost. To reduce the computational overhead, we set $E$ in the data-restricted scenario and SFT to ensure $T = 1000$ and $T = 500$ respectively.

Table 14: The effect of $K$

|  | k=1 | k=3 | k=5 | k=10 |
|---|---|---|---|---|
| Perplexity | 29.57 | 28.31 | 28.07 | 28.05 |

Table 15: The effect of $E$

|  | E=1 | E=5 | E=10 |
|---|---|---|---|
| Perplexity | 27.99 | 28.05 | 28.56 |

Table 16: Downstream evaluation after training on SlimPajama in the data-restricted scenario.

| Method | ARC-C | ARC-E | BoolQ | HellaSwag | LAMBADA | PiQA | WinoGrande | Average |
|---|---|---|---|---|---|---|---|---|
| Uniform | **25.76** | 19.58 | 60.70 | 18.62 | 9.88 | **45.43** | 50.04 | 32.85 |
| DoReMi | 21.36 | 19.58 | 60.61 | 11.66 | 6.44 | 42.94 | 49.57 | 31.06 |
| DoGE | 22.03 | 24.87 | 51.35 | 18.50 | 10.07 | 44.45 | 47.28 | 31.39 |
| Skill-it | 18.98 | **26.10** | 50.82 | 24.99 | **11.47** | 43.36 | 50.59 | 31.40 |
| Aioli | 20.34 | 19.58 | **61.71** | 19.59 | 10.89 | 44.18 | 49.41 | 32.41 |
| TANDEM | 20.34 | 20.81 | 57.55 | **25.59** | 11.20 | 40.53 | **51.07** | **32.92** |

Table 17: The effectiveness of the proxy model

| Method | Data Abundant (Perpl.) | Data Restricted (Perpl.) | SFT (Acc.) |
|---|---|---|---|
| Uniform | 25.74 | 31.53 | 74.72 |
| Online (Proxy) | 25.72 | 29.63 | 74.49 |
| Two-Stage (default) | 25.43 | 28.07 | 75.03 |

## F.4 Downstream Tasks

We also evaluate the 160M model pre-trained with SlimPajama data on ARC-C, ARC-E, BoolQ, HellaSwag, LAMBADA, PiQA and WinoGrande using the Language Model Evaluation Harness [8]. From Table 16, we see that TANDEM outperforms all the baselines, validating its effectiveness.

## F.5 The Effectiveness of the Proxy Model

It might be expected that the online-trained proxy model could also perform well. We evaluate the proxy model $u$, which has been trained with adaptive domain weights $\alpha^{(t)}$ in all the three scenarios, and compare it to the default two-stage (DMO and then train) trained model. Counter-intuitively, the performance of the online trained model falls behind the default two-stage trained model as well as the uniform baseline (Table 17). This result coincides with the findings in previous works, e.g., DoReMi and DoGE. We hypothesize that the frequent change of data distribution deteriorates the training process.

## F.6 A priori Trained Reference model Restricts DoReMi

In the experiments, DoReMi [37] doesn't perform well in the data-restricted scenarios. We hypothesize that the performance of DoReMi is restricted by the a priori trained reference model. Following previous works, we train the reference model on training data obtained using the Uniform strategy. This reference model, however, is not well-suited for the data-restricted scenario, as the many small domains overfit and cannot provide faithful signals for DMO. For a more comprehensive comparison, we train another reference model with the natural data mixture ratio (See Figure 4, the data statistics). This setting ensures no severe overfitting happens. The result in Table 18 validates our hypothesis.

# G Visualization of the Optimized Mixture Ratios.

We visualize the mixture ratio learned in each application scenario in Figure 10, Figure 11 and Figure 12. For the baselines, the average proportion over the entire training trajectory is taken. While for TANDEM, we use the average mixture ratio at the last 10% training trajectory.

Table 18: DoReMi is restricted by the a priori trained reference model

| Method | 160M | 410M | 1B |
|---|---|---|---|
| Uniform | 31.53 | 29.59 | 29.91 |
| Natural | 30.97 | 27.82 | 27.30 |
| DoReMi-Natural | 30.83 | 27.26 | 26.07 |
| TANDEM | **28.07** | **25.00** | **24.35** |

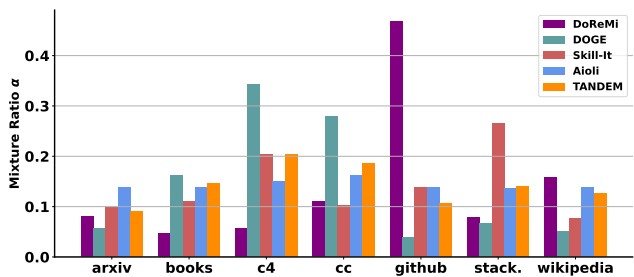

Figure 10: Mixture ratio learned by different methods in the data-abundant pretraining.

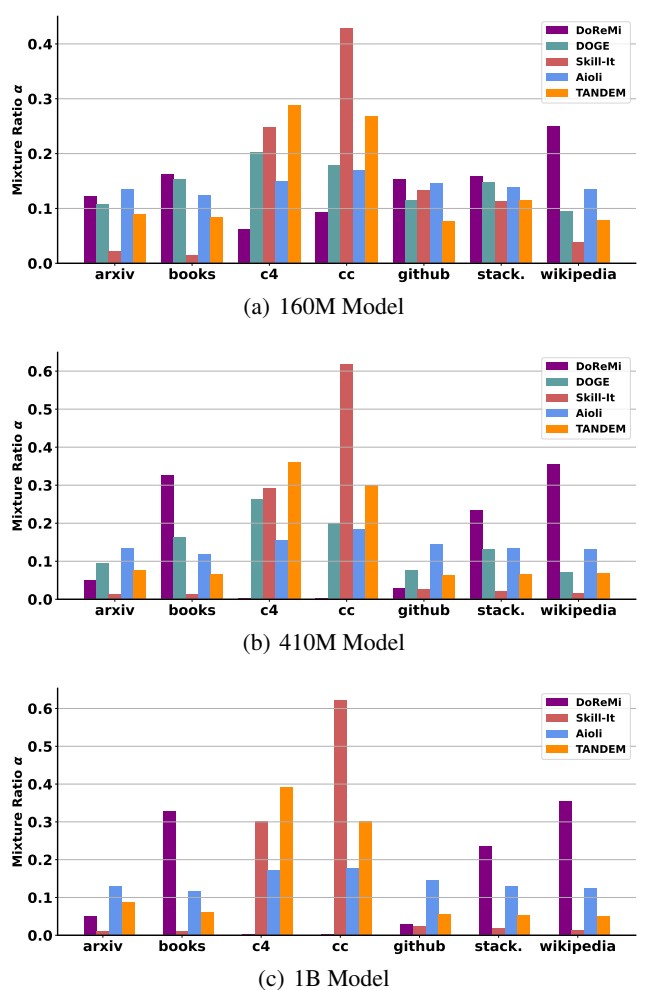

(a) 160M Model

(b) 410M Model

(c) 1B Model

Figure 11: Mixture ratio learned by different methods in the data-restricted pretraining.

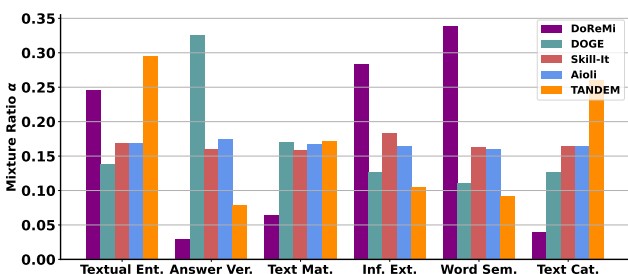

Figure 12: Mixture ratio learned by different methods in the supervised fine-tuning.

# H   Summary of Notations

Table 19: Summary of the notations used throughout this paper. Variables only used in theoretical analysis are grayed for better readability.

| Topic | Notation | Explanation |
|---|---|---|
| Data Sets | $M$ | The number of domains. |
| | $\mathcal{D}_m$ | The $m$-th domain data. |
| | $\mathcal{D}_{\text{train}}$ | Train set. |
| | $\mathcal{D}_{\text{val}}$ | Validation set. |
| Models & Parameters | $\boldsymbol{u}$ | Parameters of the proxy model. |
| | $\boldsymbol{w}$ | Parameters of the reference model. |
| | $\boldsymbol{w}^*$ | Optimal solution for the lower level problem. |
| | $\boldsymbol{w}_\gamma^*$ | Optimal solution for the penalized problem. |
| | $\boldsymbol{S}^*(\boldsymbol{\alpha})$ | Solution set for the lower level problem. |
| | $\boldsymbol{S}_\gamma^*(\boldsymbol{\alpha})$ | Solution set for the penalized problem. |
| | $\boldsymbol{\alpha}$ | Data mixture ratio |
| | $\mathcal{A}$ | The probability simplex. |
| Problems & Losses | $\mathcal{L}_{\text{train}}^m$ | Train loss on the $m$-th domain. |
| | $\mathcal{L}_{\text{val}}^m$ | Validation loss on the $m$-th domain. |
| | $\mathcal{L}_{\text{train}}$ | Overall train loss weighted by the mixture ratio $\boldsymbol{\alpha}$. |
| | $\mathcal{L}_{\text{val}}$ | Overall validation loss. |
| | $\mathcal{H}(\boldsymbol{\alpha})$ | The upper-level loss with the lower-level problem optimized. |
| | $\mathcal{H}_\gamma(\boldsymbol{\alpha}, \boldsymbol{w})$ | The loss of the penalized problem. |
| Function Properties | $\mu$ | PL coefficient of the lower-level problem. |
| | $\mu_\gamma$ | PL coefficient of the penalized problem. |
| | $L$ | Lipschitz constant for $\nabla_{\boldsymbol{w}} \mathcal{L}_{\text{train}}(\boldsymbol{\alpha}, \boldsymbol{w})$ on $\boldsymbol{w}$. |
| | $B$ | Lipschitz constant for $\mathcal{L}_{\text{train}}(\boldsymbol{\alpha}, \boldsymbol{w})$ and $\mathcal{L}_{\text{val}}(\boldsymbol{w})$ on $\boldsymbol{w}$. |
| | $\lambda$ and $\rho$ | Hessian $\nabla_{\boldsymbol{ww}} \mathcal{L}_{\text{train}}(\boldsymbol{\alpha}, \boldsymbol{w}) \succeq \lambda$ and $\nabla_{\boldsymbol{\alpha w}}^2 \mathcal{L}_{\text{train}}(\boldsymbol{\alpha}, \boldsymbol{w}) \preceq \rho$ |
| | $H$ | Lipschitz constant for $\nabla_{\boldsymbol{\alpha w}}^2 \mathcal{L}_{\text{train}}(\boldsymbol{\alpha}, \boldsymbol{w})$ and $\nabla_{\boldsymbol{ww}}^2 \mathcal{L}_{\text{train}}(\boldsymbol{\alpha}, \boldsymbol{w})$. |
| | $D$ | Upper bound for the train/validation loss. |
| | $L_\gamma$ | Lipschitz constant for $\nabla_{\boldsymbol{\alpha}} \mathcal{H}_\gamma(\boldsymbol{\alpha}, \boldsymbol{w}_\gamma^*(\boldsymbol{\alpha}))$. |
| Train | $t$ | Mixture ratio $\boldsymbol{\alpha}$ training step |
| | $T$ | The total number of $\boldsymbol{\alpha}$ update. |
| | $k$ | $\boldsymbol{u}$ and $\boldsymbol{w}$ update step. |
| | $e$ | Free $\boldsymbol{u}$ update step. |
| | $K$ | The number of $\boldsymbol{u}, \boldsymbol{w}$ probing update for one $\boldsymbol{\alpha}$ update. |
| | $E$ | The number of $\boldsymbol{u}$ free update. |
| | $\eta_{\boldsymbol{\alpha}}$ | The learning rate on $\boldsymbol{\alpha}$. |
| | $\eta_{\boldsymbol{u}}$ | The learning rate on $\boldsymbol{u}$. |
| | $\eta_{\boldsymbol{w}}$ | The learning rate on $\boldsymbol{w}$. |
| | $\gamma$ | The penalty strength. |

# I   Limitations and Future Works

Despite the advancements introduced in this work, several challenges remain open for future research. The limitations of this paper are as follows: (1) Due to limited computational resources, our experiments were conducted using models of up to 3 billion parameters. Although our experimental results demonstrate TANDEM's effectiveness at this scale, the constraint on model size limits our ability to verify whether our findings generalize to significantly larger models, such as those with 405 billion parameters. (2) Our current data mixture optimization (DMO) approach is validated on coarse, naturally occurring domain splits. For example, the SlimPajama corpus consists of seven domains sourced from different origins. The impact of using more fine-grained and intentionally designed domain splits on DMO performance remains unexplored and presents an interesting direction for future investigation.

