# OpenReview forum: "TANDEM: Bi-Level Data Mixture Optimization with Twin Networks"
_NeurIPS.cc/2025/Conference — NeurIPS 2025 poster_

### Official Review · Reviewer_SkSi · 2025-06-26

**Clarity:** 2
**Significance:** 2
**Originality:** 2
**Rating:** 4
**Confidence:** 4

**Summary:**

This paper introduces TANDEM, an algorithm designed to optimize the domain-specific mixture ratios of training data for LLMs. The authors frame data mixture optimization (DMO) as a bi-level optimization problem and reformulate it into a single-level penalized objective.
TANDEM solves this by employing "twin networks": a proxy model trained on training data and a reference model trained on both training and validation data. By the difference in losses between these twin models, the method up-weights domains that show more significant improvement from additional validation data.
Extensive experiments across data-abundant pretraining, data-restricted pretraining, and supervised fine-tuning scenarios validate TANDEM's effectiveness and superiority over existing methods.

**Questions:**

1. Why not treat the proposed algorithm as an online method? Since proxy model $u$ or reference model $w$ has been trained with adaptive domain weights $\alpha^t$, is it possible to train a base model using procedure of TANDEM directly? (Treat $u$ or $w$ as final LM.) I'm curious the performance of them.
2. One recent work, Chameleon [1], also considers data mixing in finetuning. Could you compare TANDEM with Chameleon? That would be great if you can also show some empirical results.

[1] Xie, Wanyun, Francesco Tonin, and Volkan Cevher. "Chameleon: A Flexible Data-mixing Framework for Language Model Pretraining and Finetuning." ICML 2025.

**Ethical Concerns:**

["NO or VERY MINOR ethics concerns only"]

**Final Justification:**

- It's interesting to consider data-restricted scenarios.
- The paper expands data mixing methods into supervised fine-tuning.
- I would suggest including the performance of downstream tasks in the paper.
- Comparing computational cost with other baselines should be added.

**Limitations:**

See weakness.

**Quality:**

1

**Strengths And Weaknesses:**

**Strength**
1. The paper's re-formulation of the bi-level DMO problem into a single-level penalized objective is insightful.
2. TANDEM's dynamic updating and synchronization of both proxy and reference models address limitations of prior work.
3. It's important to consider Data-Abundant and Data-Restricted scenarios separately.
4. It's interesting to consider data mixing in supervised fine-tuning.

**Weakness**
1. The derivation from Eq (2) to Eq (4) is unclear. Does $\nabla L_{val}(\alpha, w_k)$ in Eq (4) mean the gradient on validation set is computed with domain weights $\alpha^t$ rather than uniform weights? However, shouldn't it be uniform weights according to Eq (2)?
2. The empirical results on SlimPajama only show perplexity, but it doesn't test on downstream tasks. Although average perplexity can represent model's performance, downstream tasks can show models' real capability. Many data mixing works like Doge, RegMix, Chameleon [1] show various downstream tasks' accuracy.
3. Although time of TANDEM is competitive to other data mixing methods as shown in Table 3, the computational cost is related to $K$ and $E$. Could you discuss more about computational time with respect to $K$ and $E$? In addition, could you report FLOPs during training?

---

> ### Author Rebuttal · Authors · 2025-07-31
>
> Thank you for your constructive comments. We address your concerns below:
>
> >Q1: $\nabla L _ \text{val}\left(\boldsymbol{\alpha}, \boldsymbol{w} _ k\right)$ in Equation 2 should be $\nabla L _ \text{val}\left( \boldsymbol{w} _ k\right)$.
>
> Yes, there is a typo in Equation (4). The gradient on the validation set is computed with the uniform weights. Thank you for pointing this out, we will make it clearer in the revision.
>
>
> >Q2: Performance on downstream tasks.
>
> Thanks for the suggestion, we evaluate the 160M models trained on SlimPajama on ARC-C, ARC-E, BoolQ, HellaSwag, LAMBADA, PiQA and WinoGrande using the Language Model Evaluation Harness [4] as in Aioli [1]. The result is as follows:
> | Method   | ARC-C | ARC-E | BoolQ | HellaSwag | LAMBADA | PiQA | WinoGrande | Average |
> |:--------:|:-----:|:-----:|:-----:|:---------:|:-------:|:----:|:----------:|:-------:|
> | Uniform  | **25.76** | 19.58 | 60.70 | 18.72     | 9.98    | **45.43**| 50.04      | 32.85   |
> | DoReMi   | 21.36 | 19.58 | 60.61 | 11.66     | 6.44    | 42.94| 49.57      | 31.06   |
> | DoGE     | 22.03 | 24.87 | 51.35 | 18.50     | 10.07   | 44.45| 47.28      | 31.39   |
> | Skill-it | 18.98 | **26.10** | 50.82 | 24.99     | **11.47**   | 43.36| 50.59      | 31.40   |
> | Aioli    | 20.34 | 19.58 | **61.71** | 19.59     | 10.89   | 44.18| 49.41      | 32.41   |
> | TANDEM   | 20.34 | 20.81 | 57.55 | **25.59**     | 11.20   | 40.53| **51.07**      | **32.92**   |
>
> We see that TANDEM outperforms all the baselines, though not as significantly as on the perplexity. Actually, as many recent works[1][5][6][7] pointed out, lower perplexity is not necessarily positively correlated with higher performance on downstream tasks.
> This disparity might be caused by the distribution shift between the pre-training data and the downstream evaluation task. Recall that the bilevel formulation directly optimizes the model performance on validation data, which is, in our case also the pre-training data instead of the downstream tasks. When an algorithm is completely unaware of the downstream tasks,  we do not expect it to universally outperform the others. We do believe that by establishing a comprehensive and diverse validation system, TANDEM will significantly improve the development of LLM. We will add an in-depth discussion on this in the revision.
>
> >Q3:  Computational time with respect to $K$ and $E$. FLOPs during training.
>
> We conduct a computational complexity analysis contrasting TANDEM and vanilla model training on the same dataset. The complexity is as follows:
>
> |  Method | Computational Complexity |
> |:---------:|:---:|
> | Vanilla Model Train|$C T (E + K)$ |
> | TANDEM             |$C T (E + K) +  C  T  K + \frac{2}{3}C T$ |
>
> $C$ is the complexity of one step training of an ordinary model. $T$ is the update number of $\boldsymbol{\alpha}$. $K + E$ compose the mixture ratio update interval ($K$ steps of probing, and $E$ steps of free $\boldsymbol{u}$ training). In this way, training on $\boldsymbol{u}$ takes $C T (E + K) $ computation, additional update of $\boldsymbol{w}$ takes $ C  T  K $, and update on $\boldsymbol{\alpha}$ takes $\frac{2}{3}C T$ (forward process of two models $\boldsymbol{u}$ and $\boldsymbol{w}$, here we simply assume backward takes 2$\times$ computation as the forward process). Please refer to Figure 1 for a better understanding. Compared to vanilla model training on the same dataset~(train only $\boldsymbol{u}$ for $T(K+E)$ steps), which takes $C  T  (K+E)$ computation, TANDEM only introduce $C  T  K + \frac{2}{3}C T$ additional computation, less than 2$\times$ vanilla training in most cases. This complexity dominates the FLOPs during training. We will include FLOPs in the experiments in the revision.
>
> >Q4:  Why not establish an online method and treat $\boldsymbol{u}$ or $\boldsymbol{w}$ as the final model?
>
> Thank you for the suggestion. We evaluate the model $\boldsymbol{u}$ which has been trained with adaptive domain weights $\boldsymbol{\alpha^{t}}$ in all the three scenarios ($\boldsymbol{w}$ is (re)initialized with $\boldsymbol{u}$ periodically so we consider just $\boldsymbol{u}$), and compare it to the default two-stage (DMO and then train) trained model.
> | Method             | Data Abundant (Perpl.) | Data Restricted (Perpl.) | SFT (Acc.) |
> |:------------------:|:---------------------:|:-----------------------:|:---------:|
> | Uniform            |         25.74         |          31.53          |   74.72   |
> | Online             |         25.72         |          29.63          |   74.49   |
> | Two-Stage (default)|         25.43         |          28.07          |   75.03   |
>
> From the table above, we see that the performance of the online trained model $\boldsymbol{u}$ falls behind the default two-stage trained model as well as the uniform baseline. This result coincides with the findings in previous works, e.g., Table 6 in DoReMi(The proxy model underperforms the baselines) and Figure 8 in DoGE. We hypothesize that the frequent change of training data distribution deteriorates the training process. Though TANDEM in its current form is not suitable to be used as an online method, we totally agree that establishing online methods remains an important future work.
>
> >Q5:  One recent work, Chameleon [3], also considers data mixing in finetuning. Could you compare TANDEM with Chameleon?
>
> There might be a misunderstanding. The "finetuning" in CHAMELEON much resembles (continual) pretraining where the data used (Wiki-40 and Stack-decup) are also unlabeled plain texts and consumed only once, while in our SFT setting we focus on the "prompt \& answer" pair data, which usually induces much noisy gradients, allows multi-epoch training and suffer from overfitting problems.
> Similar to Skill-it, CHAMELEON determines the optimal mixture ratio according to a pre-computed fixed $M \times M$ domain correlation matrix ($A$ in skill-it and $XX^T$ in CHAMELEON), which encodes the data efficacy of each domain. However, this kind of method fades when data in certain domains quickly depletes and upsampling kicks in (our data-restricted scenario) because the data efficacy changes, the domain correlation matrix is no longer aligned with the current data characteristics.
>
> Finally, note that CHAMELEON is released on May 30, 2025, after the submission deadline of NeurIPS 2025, so by no means at that time can we compare CHAMELEON with TANDEM. Due to the limited rebuttal duration, we defer this comparison to the revision.
>
> We thank the reviewer again for the review. If you have any more questions about our replies, or any further questions or concerns, please let us know.
>
> [1] Mayee F. Chen, Michael Y. Hu, Nicholas Lourie, Kyunghyun Cho, and Christopher Ré. Aioli: A unified optimization framework for language model data mixing. In International Conference on Learning Representations, 2025.
>
> [2] Mosh Levy, Alon Jacoby, and Yoav Goldberg. Same task, more tokens: the impact of input length on the reasoning performance of large language models. In Annual Meeting of the Association for Computational Linguistics.
>
> [3] Xie, Wanyun, Francesco Tonin, and Volkan Cevher. Chameleon: A Flexible Data-mixing Framework for Language Model Pretraining and Finetuning. In International Conference on Machine Learning, 2025.
>
> [4] Leo Gao, Jonathan Tow, and Baber Abbasi et.al. A framework for few-shot language model evaluation, 2024.
>
> [5] Hong Liu, Sang Michael Xie, Zhiyuan Li, and Tengyu Ma. Same pre-training loss, better downstream: Implicit bias matters for language models. In International Conference on Machine Learning 2023,
>
> [6] Yi Tay, Mostafa Dehghani and Samira Abnar. Scaling laws vs model architectures: How does inductive bias influence scaling? In Conference on Empirical Methods in Natural Language Processing, 2023.
>
> [7] Mengzhou Xia, Mikel Artetxe and Chunting Zhou et.al.. Training trajectories of language models across scales. In Association for Computational Linguistics, 2023.

---

> > ### Comment · Reviewer_SkSi · 2025-08-04
> >
> > Thanks for the authors' detailed responses.
> >
> > The authors claim that "TANDEM's computational cost is usually less than 2$\times$ vanilla training.". It's a bit hard to say if it's an acceptable cost. I suggest adding a comparison of other baselines in the paper.
> >
> > My other concerns have been addressed. I would like to raise my score.

---

> ### Author Response · Authors · 2025-08-05
> **Response to Reviewer SkSi's Comment**
>
> Dear reviewer,
>
> We are delighted to hear that we were able to address your concerns.
>
> We compare the computational complexity of different baselines:
> |  Method | Computational Complexity |
> |:---------:|:---:|
> | Vanilla Model Train|$C T (E + K)$ |
> |DoReMi| $\frac{8}{3}C T (E+K)$|
> |DoGE | $2CT(E+K) + M T(E+K) N$|
> |Skill-it| $(\frac{M}{4} + 1)CT(E+K) + \frac{M}{3}CT $ |
> |Aioli| $CT(E+K) + MCTK + \frac{1}{3}M^2C T $|
> | TANDEM             |$C T (E + K) +  C  T  K + \frac{2}{3}C T$ |
>
> $N$ is the number of parameters. The overall computational cost of these methods is of the order: Aioli > Skill-it > DoReMi  $\approx$ DoGE $\approx$ TANDEM. (Skill-it requires training $M$ models with data from each domain for $H$ steps in advance, we use $H = \frac{1}{4}T(K+E)$ in our analysis).  Note that though the complexity of DoGE is not particularly large, it requires per-domain gradient computation, which is unfriendly to the computing kernel and takes much more time. For TANDEM, our current implementation includes $T$ times model copying and loading (setting $\boldsymbol{w}=\boldsymbol{u}$ , line 102 ), incuring non-negligible time-cost. A more optimized implementation can greatly reduce the time consumed by TANDEM. We will add detailed computational complexity analysis in the revision.
>
> As for CHAMELEON, it is the most efficient, as it requires only vanilla training ($<C T (E+K)$) to get the "domain correlation". Nevertheless, as we mentioned in the rebuttal, CHAMELEON is not able to capture the data efficacy change in the data-restricted scenario. We compare CHAMELEON and TANDEM. For the 160M/410M/1B model, CHAMELEON gets perplexity 29.85, 27.65, and 27.55, higher than TANDEM's 28.07, 25.0, and 24.35.
>
> Once again, thank you for your valuable feedback and kind consideration.
>
> Best regards,
>
> The authors

---

### Official Review · Reviewer_Vq6v · 2025-06-29

**Clarity:** 3
**Significance:** 3
**Originality:** 2
**Rating:** 5
**Confidence:** 4

**Summary:**

This paper introduces TANDEM, a novel method for data mixture optimization in the context of large language model (LLM) training. The authors formulate the optimization of domain-specific mixture ratios as a bi-level optimization problem, and transform it into a penalized single-level formulation. They propose a twin-network architecture where a proxy model is trained on primary data, and a reference model is dynamically trained with additional data. The difference in loss between these two models serves as a signal to reweight the domain mixture. TANDEM is theoretically grounded, enjoys convergence guarantees, and is empirically validated across three representative scenarios: data-abundant pretraining, data-restricted pretraining, and supervised fine-tuning.

**Questions:**

1. Scalability to LLM-scale models (10B+ parameters): While 160M to 1B models are explored, can TANDEM scale to 10B+ scale? Are there practical bottlenecks (e.g., memory, wall-clock time) for training two models in parallel at scale?
2. Downstream Generalization: The current evaluation is focused on perplexity or loss. Have you tested whether mixture ratios learned by TANDEM improve generalization in downstream tasks, especially zero-shot / few-shot ability in different benchmarks for pretraining models (such as MMLU, GSM8K)?
3. The authors provide a useful analysis on how increasing K reduces gradient variance (Fig. 8). However, it remains unclear how sensitive the final model performance (e.g., perplexity, accuracy) is to different values of K, especially across different domains or tasks. Ablation studies on these dimensions, as well as the feasibility of adaptive or dynamic K, would further strengthen the claims.

**Ethical Concerns:**

["NO or VERY MINOR ethics concerns only"]

**Final Justification:**

I appreciate the authors' thorough and comprehensive rebuttal. I have carefully reviewed the responses and the additional experiments presented, which addressed most of my concerns. I revised my score accordingly.

**Limitations:**

yes

**Quality:**

3

**Strengths And Weaknesses:**

Strengths：
1. The paper is well-organized with clear motivation, rigorous derivations, and strong empirical validation.
2. The transformation of the bi-level optimization into a penalized single-level form is theoretically grounded, with convergence analysis included.
3. The method is efficient and scalable, avoiding Hessian computation, and is applicable to a wide range of scenarios, especially under-explored settings like SFT.
4. Extensive experiments in multiple scenarios (pretraining, low-resource, SFT) show TANDEM outperforming strong baselines.

Weakness：
1. Although the method is presented as novel, it closely builds on DoReMi and DoGE. The conceptual contribution lies more in unifying and refining rather than innovating from scratch.
2. The paper lacks discussion on how TANDEM could be used in industry-scale LLM training pipelines and what challenges may arise there.
3. Limited Task Diversity in SFT: The SFT experiments focus on only six tasks with small datasets; broader evaluation would strengthen generalizability claims.

---

> ### Author Rebuttal · Authors · 2025-07-31
>
> Thank you for your constructive comments. We address your concerns below:
>
> >Q1: Although the method is presented as novel, it closely builds on DoReMi and DoGE.
>
> While TANDEM does draw inspiration from prior work like DoReMi and DoGE, it represents significant conceptual and methodological advancements in several key aspects:
>
> **Theoretical Foundation:**  Unlike DoReMi and DoGE's heuristic approach (group DRO \& alternating model parameters $\boldsymbol{w}$ and mixture weights $\boldsymbol{\alpha}$), TANDEM establishes a rigorous bi-level optimization framework with provable convergence guarantees (Theorem 1), representing a fundamental theoretical contribution.
>
> **Architectural Innovation:**  The twin-network design (Figure 1) introduces novel synchronization mechanisms between proxy and reference models, addressing critical limitations: Dynamic reference model vs DoReMi's static reference model. Light-weight loss comparison vs  DoGE's cumbersome and noisy per-domain gradient estimation. The proposed method is elegant and conceptually appealing, as the reviewer jR7t stated.
>
> **Valuable Field Insights:**   As the reviewer jR7t recognized, this paper goes beyond the algorithmic design to discuss when and why DMO matters. Providing valuable insights to the DMO field.
>
> The synthesis of these elements - theoretical grounding, architectural innovation, and valuable field insights - constitutes innovation beyond simply unifying existing approaches.
>
>
> >Q2:  How TANDEM could be used in industry-scale LLM training pipelines and what challenges may arise there.
>
> TANDEM can be used to construct the final training set after samples in each domain are properly pre-processed and de-duplicated, aiming at building a well-performing and balanced model.
>
> As the reviewer pointed out, directly applying TANDEM to industry-scale LLM poses significant challenges. Maintaining the reference model and proxy model simultaneously requires nearly doubled memory. One potential bypass is to use smaller models for data mixture optimization and transfer the learned mixture ratio to LLM training. Previous works like DoReMi and DoGE have demonstrated the effectiveness of this transfer. This also applies to TANDEM. In Figure 6, we show that the optimal mixtures learned with models of different scales are quite consistent. So the $\boldsymbol{\alpha}$ learned with a smaller model can be safely applied to a larger model train.
>
> >Q3: Limited Task Diversity in SFT with small datasets.
>
> Thanks for your suggestion. We have already tested TANDEM on tasks of different formats, including open-ended text generation, multiple choice, and True/False tasks. Here we further increase the data scale of the SFT experiments. We use 6 major categories (containing 99 tasks) from Natural Instructions, each contains tasks of different formats. The statistics are as follows:
> | Method               | Num. Sample |
> |:--------------------:|:-----------:|
> | Textual Entailment   | 79332       |
> | Answer Verification  | 13195       |
> | Text Matching        | 47297       |
> | Information Extraction | 31053    |
> | Word Extraction      | 17294       |
> | Text Categorization  | 89572       |
>
> We use the same hyperparameters as in section 4.1. The comparison on this larger SFT dataset is as follows:
> | Method   | Text Ent. | Ans Ver. | Text Mat. | Inf Ext. | Word Sem. | Text Cat. | Avg Metric $\uparrow$ | Test Loss $\downarrow$ |
> |:--------:|:---------:|:--------:|:---------:|:--------:|:---------:|:---------:|:-----------:|:----------:|
> | Uniform  | 84.4    | 75.1 | 86.2  | 77.8   | 88.3 | 83.2 | 82.5 |  0.231       | 0.231      |
> | DoReMi   | 85.3      | 74.9     | 83.6      | 78.1     | 87.0      | 79.0      | 81.6        | 0.249      |
> | DoGE     | 81.4      | 76.7     | **86.5**  | **78.6** | 88.3      | 82.4      | 82.4        | 0.297      |
> | Skill-It | 84.1      | 75.0     | 86.4      | 78.3     | 87.9      | 83.7      | 82.6        | 0.232      |
> | Aioli    | 83.9      | 75.8     | 86.2      | 78.0     | 88.3      | 84.0      | 82.7        | 0.229      |
> | TANDEM   | **85.3**  | **76.3** | 86.2      | **78.6** | **88.5**  | **84.9**  | **83.3**    | **0.208**  |
>
> From the table we see that TANDEM still outperforms all the baselines on this much larger SFT dataset.
>
> >Q4: Experiments on larger models.
>
> Thanks for the suggestion, we’ve added 3B scale experiments. Please kindly refer to our response to question 2 for reviewer 3KfP.
>
> > Performance on downstream tasks.
>
> Thanks for the suggestion, we've added downstream few shot experiments and indepth discussion. Please kindly refer to our response to the Q2 for reviewer SkSi. MMLU & GSM8K are too hard for models and data of this size, so we evaluate ARC-C, ARC-E, BoolQ, HellaSwag, LAMBADA, PiQA and WinoGrande as in Aioli
>
> > Sensitivity of the final model performance on $K$. The feasibility of adaptive or dynamic $K$
>
> Thanks for your suggestion. Here we add a sensitivity analysis on $K$, aiming at inspecting how vary $K$ will affect the final model performance. We test $K=1,5,10$ in the data-restricted scenario, and the result is as follows:
> |           | K=1  | K=3  | K=5 | K=10 |
> |:---------:|:----:|:----:|:----:|:----:|
> | Perplexity|29.57 |28.31 |28.06 |28.05 |
>
> From this table, we see that too small $K$ may result in less fidel $\boldsymbol{\alpha}$ update direction, thus sub-optimal data mixture ratio and higher perplexity. When $K$ is large enough for $\boldsymbol{\alpha}$ update probing,   increasing $K$ will not induce further benefits.
>
> For adaptive or dynamic $K$, we agree that this can be an interesting and fruitful direction. We left it for future work.
>
>
> We thank the reviewer again for the review. If you have any more questions about our replies, or any further questions or concerns, please let us know.

---

> > ### Author Response · Authors · 2025-08-04
> > **Thank you again for your review.**
> >
> > Dear reviewer:
> >
> > Thank you again for your time reviewing our paper. We would appreciate it if you could confirm that our responses address your concerns. We would also be happy to engage in further discussions to address any other questions that you might have.
> >
> > Best regards

---

> > ### Comment · Reviewer_Vq6v · 2025-08-05
> > **reviewer's comment to authors' rebuttal**
> >
> > I appreciate the authors' thorough and comprehensive rebuttal. I have carefully reviewed the responses and the additional experiments presented, which addressed most of my concerns.
> > I will revise my score accordingly.

---

> > > ### Author Response · Authors · 2025-08-06
> > >
> > > Dear reviewer,
> > >
> > > We are delighted to hear that we were able to address most of your concerns. We will incorporate this valuable feedback and include the larger-scale experiments, discussion on applying in industry-scale LLMs, more diverse large-scale SFT, performance on downstream tasks, and the sensitivity analysis on $K$, in the revised version of our paper.
> > >
> > > Once again, thank you for your valuable feedback and kind consideration for revising the score.
> > >
> > > Best regards,
> > >
> > > The authors

---

### Official Review · Reviewer_jR7t · 2025-07-01

**Clarity:** 2
**Significance:** 3
**Originality:** 4
**Rating:** 3
**Confidence:** 4

**Summary:**

The paper proposes TANDEM, a new algorithm for DMO in LLM training. It reformulates the bi-level optimization of domain mixture ratios into a single-level penalized problem, which is then solved using a twin-network approach: a proxy model trained on primary data and a reference model trained on additional validation data. TANDEM measures the impact of data from different domains by comparing these models, dynamically up-weighting domains that improve performance. TANDEM offers better stability, theoretical convergence guarantees, and strong empirical results—especially in underexplored settings like supervised fine-tuning and data-restricted regimes.

**Questions:**

1. Please provide a more detailed explanation of how Equation (5) is derived. The current presentation is abrupt and lacks sufficient intermediate steps for readers to follow.

2. One of your central claims is that “data mixture optimization is particularly beneficial in scenarios with limited data availability rather than traditional pretraining setups with abundant domain data.” However, the last paragraph of Section 2.4 is mostly high-level speculation. It would be more convincing if you could include either a theoretical extension (e.g., following from Proposition 1) or empirical evidence that directly supports this claim.

3. Many hyperparameters vary across scenarios (e.g., different values of $K$, $E$, and even context length). Could you elaborate on the rationale behind these choices? Is TANDEM’s performance robust to such variations?

4. You argue that uniform mixing performs well in the first (data-abundant) scenario. However, even in the last two scenarios, DoReMi and DoGE consistently underperform the uniform baseline. I’m concerned whether these baselines were properly implemented and tuned.

5. In the analysis of the probing step $K$, why do you fix $\alpha$? Doesn’t this deviate from how the actual algorithm works?

6. (Not affecting the score):
(1) Some notations (e.g., $\prod_A$) should be better explained.
(2) In Line 155, it is misleading to refer to cosine similarity directly as “variance.”
(3) For better clarity, please annotate all figures and tables (e.g., Table 3, Figures 5 and 6) with the corresponding scenario (data-abundant, data-restricted, or SFT).

**Ethical Concerns:**

["NO or VERY MINOR ethics concerns only"]

**Final Justification:**

I believe the presentation of the paper has improved substantially after the rebuttal, and many of the previously qualitative arguments are now sufficiently supported. Therefore, I have raised my score. However, the use of relatively small models (understandable given the computational cost) and the statistical significance of some experimental results (e.g., Tables 2 and 4) remain concerns.

**Limitations:**

yes

**Paper Formatting Concerns:**

None.

**Quality:**

2

**Strengths And Weaknesses:**

**Strengths**
1. The core idea of reformulating DMO as a single-level penalized problem and solving it via twin networks is elegant and conceptually appealing.

2. I especially appreciate Sections 2.3 and 2.4, which clearly connect TANDEM with prior methods (e.g., DoReMi and DoGE), offering a unified view that highlights both similarities and improvements, and go beyond algorithmic design to discuss **when and why** DMO matters.

**Weaknesses**
1. The methodology section lacks clarity in several places, and many claims are made without sufficient justification or analysis. For example, (1) in Section 2.2, the derivation of Equation (5) could be explained more clearly; (2) the final paragraph of Section 2.4 reads more like speculation, with no analytical or empirical support and no follow-up on Proposition 1.

2. The experimental setup relies on models that are too small to draw strong conclusions. Most experiments are done with models <1B, with only one section using a 1B model. While I understand the computational cost of DMO, these models are not sufficiently representative. It is also unclear why the SFT experiments use only a 0.5B model—does it really impose a high cost?

3. The experimental results raise several concerns: (1) Table 4 does not report any statistical significance, and the observed differences appear too subtle to be convincing; (2) in all settings, several baselines (e.g., DoReMi and DoGE) perform worse than uniform, which is surprising and concerning. While the paper argues that uniform mixing works well in data-abundant settings, the poor performance of baselines in the other two scenarios warrants deeper investigation.

4. The writing quality is unprofessional, with numerous grammatical and typographical errors throughout the paper, including L62, L99, L111, L115, L149, L151, and the caption of Table 2.

----

I would be happy to raise my score if the authors can adequately address the weaknesses and questions I raised.

---

> ### Author Rebuttal · Authors · 2025-07-31
>
> Thank you for highlighting the strengths of our method—its elegance, comprehensive scope, and its deeper discussion of when and why DMO matters. We believe this provides meaningful insights into the DMO problem. We address your concerns below:
>
> >Q1: The derivation of Equation (5) could be explained more clearly.
>
> In the objective (2), only the $\gamma (\mathcal{L} _ {\text{train}}( \boldsymbol{\alpha}, \boldsymbol{w})- \min_{\boldsymbol{u}} \mathcal{L}_{\text{train}}(\boldsymbol{\alpha}, \boldsymbol{u}))$ is related to $\boldsymbol{\alpha}$. To see how this objective induces the update rule Equation (5), we first inspect the training loss. By definition, it is the $\boldsymbol{\alpha}$ weighted domain-wise loss: $\mathcal{L} _ {\text{train}}(\boldsymbol{\alpha}, \boldsymbol{w}) = \sum _ {m=1}^M \boldsymbol{\alpha} _ m\mathcal{L} _ {\text{train}}^{m}(\boldsymbol{w})$. The gradient w.r.t $\boldsymbol{\alpha}$ is exactly the domain-wise loss: $\nabla _ {\boldsymbol{\alpha}}\mathcal{L} _ {\text{train}}(\boldsymbol{\alpha}, \boldsymbol{w}) = \mathcal{L} _ {\text{train}}^{1:M}(\boldsymbol{\alpha}, \boldsymbol{w})$. After the $K$ steps update of $\boldsymbol{u}$ and $\boldsymbol{w}$ (Equation (3) (4)), substitute $\boldsymbol{u}_k$ and $\boldsymbol{w}_k$ results in the update rule Equation (5). $\Pi _ {\mathcal{A}}(\cdot)$ projects the updated $\boldsymbol{\alpha}$ into a probabilistic simplex. Running equation (3), (4) and (5) gives the solution of the bilevel DMO solution as shown in Theorem 1. We will make this clearer in the revision.
>
> >Q2: The final paragraph of Section 2.4 reads more like speculation.
>
> There might be a misunderstanding. This section is not speculation. Let us elucidate the rationale underlying the description.
> In Proposition 1, we see that as long as the per-domain generalization gap $ | \mathcal{L} _ {\text{val}}^m - \mathcal{L} _ {\text{train}}^m | \rightarrow 0$, uniform weighting emerges as a valid solution. But why in the conventional data-abundant scenario do the generalization gaps tend to 0? Note that for uniform $\boldsymbol{\alpha}$, the training data and validation data are independently identically distributed (See Equation 1, $\mathcal{L} _ {\text{val}}$ always takes uniform $\boldsymbol{\alpha}$). The validation loss can be used as a proxy for population risk. Recall that in empirical risk minimization, mini-batch stochastic gradient follows the gradient of the true population risk as long as no examples are repeated ([1], Page 277), so the $\mathcal{L} _ {\text{val}}^m$ and  $\mathcal{L} _ {\text{train}}^m$ decrease at a similar speed during the training when all the training data are consumed only once (the first epoch), that is, our conventional data-abundant scenario. So $|\mathcal{L} _ {\text{val}}^m - \mathcal{L} _ {\text{train}}^m| \rightarrow 0$ in this case. In the data-restricted scenario (In a general sense, including cases where the overall data volume is large, but certain domains are relatively small), the situation changes. Samples from smaller domains are inevitably upsampled, mini-batch stochastic gradient no longer follows the gradient of the population risk (the validation loss). The generalization gap $|\mathcal{L} _ {\text{val}}^m - \mathcal{L} _ {\text{train}}^m|$ increases. A typical example is overfitting on the smaller domains. In this way, the uniform $\boldsymbol{\alpha}$ is no longer a solution for the DMO problem.
>
> For the empirical evidence, from Table 2, we see that in the conventional data-abundant scenario, no one approach can significantly outperform Uniform (supported also by [2]), showing that Uniform might already be a valid solution. While in the data-restricted scenario, a number of approaches including DoGE, Skill-it, Aioli and TANDEM outperform Uniform by a large margin.
>
> >Q3: The rationale behind the choice of hyperparameters.
>
> **Context length:**  In the data-abundant pretraining scenario, we use the context length 2048, the same as in Aioli [1], so that we can compare directly with the results given in Aioli~[1]. For the data-restricted scenario and SFT, we keep a unified context length of 512 as the generated token lengths in the SFT tasks are all smaller than 512. The context length will not affect the results. In the data-restricted scenario, we test context length 2048 and TANDEM reaches a final perplexity 28.04, 25.17, 25.21 for 160M, 410M, and 1B models, which is very similar to the 512 length results.
>
> **$K$:**   $K$ is the number of probing steps used to estimate the proper $\boldsymbol{\alpha}$ update direction (Equation (3)$\sim$(5)). Larger $K$ helps suppress the gradient variance and provides more fidel $\boldsymbol{\alpha}$ update direction. Pretraining is relatively stable, so we use a smaller $K=5$ in the data-abundant and data-restricted scenarios. The gradient variance in SFT, however, is much higher (Figure 2), so we use a larger $K=20$. Furthermore, we conduct a sensitive analysis on how $K$ will affect the final model performance. The result shows that $K$ should be sufficiently large to ensure proper $\boldsymbol{\alpha}$ update, while further increasing $K$ beyond this point will not induce additional benefits. Please kindly refer to our response to question 6 for reviewer Vq6v.
>
> **$E$:**   Given the total number of training steps (amounts of data to be used), $E$ determines the number of $\boldsymbol{\alpha}$ updates during DMO: $T = \frac{total\  steps}{E}$. In the data-abundant scenario, we chose $E=20$ so that the mixture ratio is updated for $T = 2000$ steps. As one $\boldsymbol{\alpha}$ update requires an addition $K$ steps $\boldsymbol{w}$ updates, a larger $T$ means higher additional computational cost. To reduce the computational overhead, we set $E$ in the data-restricted scenario and SFT to ensure $T=1000$ and $T=200$ respectively.
>
> The value of $E$ is not deliberately tuned. Here we provide a sensitive analysis on $E$. We test $E=1,5,10$ in the 160M data-restricted scenario, and the result is as follows:
> |           | E=1  | E=5  | E=10 |
> |:---------:|:----:|:----:|:----:|
> | Perplexity|27.99 |28.05 |28.56 |
>
> TANDEM is not that sensitive to $E$ as long as it makes $T$ large enough so that $\boldsymbol{\alpha}$ is sufficiently updated.
>
> >Q4: Experiments on larger models.
>
> Thanks for the suggestion, we’ve added 3B scale experiments. Due to limited space, please refer to our response to question 2 for reviewer 3KfP.
>
> >Q5: The improvement of TANDEM in Table 4 (SFT) is not significant.
>
> In Table 4, TANDEM doesn't yield particularly strong empirical results because the final metric for the 6 tasks is inconsistent with the loss. While TANDEM significantly decreases the test loss, the performance doesn't improve proportionally.
>
> In response to question 3 for reviewer Vq6v, we conducted a much larger-scale SFT experiment. On this larger dataset, TANDEM demonstrates much significant advantages over the baselines, showing the effectiveness of the proposed method. Please kindly refer to our answer to question 3 for reviewer Vq6v.
>
> >Q5: Why are DoReMi and DoGE not as good as Uniform?
>
> As discussed in Section 2.4, Uniform is already a valid solution for the bi-level DMO problem in the data-abundant scenario, so it is hard for DoReMi and DoGE to outperform Uniform significantly. We then illustrate why the two methods suffer in the other scenarios.
>
> **DoReMi:**  The problem of DoReMi lies in that it depends on a pre-trained static reference model. Imagine that in the data-restricted scenario and SFT, the reference model overfits on certain small domains (performs extremely low training loss, so the excessive losses are high), DoReMi may incorrectly continue to up-weight the domains to reduce the corresponding excessive losses. Besides, we know that, different from pretraining where the training loss decreases steadily, in SFT, there are always sudden downward jumps in training loss when the data repeats [3]. There are also chances that the SFT loss goes even to 0 for a large model. These pose greater challenges for DoReMi to obtain a good reference model.
>
> **DoGE:**   In Table 2, we see that DoGE outperforms Uniform in the data-restricted scenario. While for SFT, DoGE suffers because it relies on the inner product of per-domain gradients, which exhibits large variance and is difficult to provide reliable $\boldsymbol{\alpha}$ update directions (Figure 8). On the contrary, our TANDEM does not rely on the gradient estimation and utilizes $K$ steps probing to get more reliable $\boldsymbol{\alpha}$ updates.
>
> >Q6: In the analysis of the probing step, why do you fix $\boldsymbol{\alpha}$?
>
> We fix $\boldsymbol{\alpha}$ to isolate the effect of $K$ in suppressing the hyper-gradient variance. Update of $\boldsymbol{\alpha}$ will introduce an extra variable--the value of the mixture ratio $\boldsymbol{\alpha}$. Nevertheless, we agree that an analysis on how $K$ directly affects the final model performance is important. We add a sensitive analysis, please kindly refer to our response to question 6 for reviewer Vq6v.
>
>
> >Q7:  Typos, grammatical, typographical issues and other suggestions helps improving the clarity.
>
> Thanks very much for your careful checking, we've fixed these typos and thoroughly polished the revision.
>
> Since we have replied to all the questions, if you find our answers satisfactory, we respectfully ask that you please consider increasing the score. If you have any more questions or concerns, please let us know and we will be happy to answer.
>
>
>
>
> [1] Ian Goodfellow, Yoshua Bengio, and Aaron Courville. Deep Learning, MIT Press, 2016
>
> [2] Mayee F. Chen, Michael Y. Hu, Nicholas Lourie, Kyunghyun Cho, and Christopher Re. Aioli: A unified optimization framework for language model data mixing. In International Conference on Learning Representations, 2025.
>
> [3] Jeremy Howard and Jonathan Whitaker. Can LLMs learn from a single example? 2023

---

> ### Author Response · Authors · 2025-08-04
> **Thank you again for your review.**
>
> Dear reviewer:
>
> Thank you again for your time reviewing our paper. We would appreciate it if you could confirm that our responses address your concerns. We would also be happy to engage in further discussions to address any other questions that you might have.
>
> Best regards

---

> > ### Comment · Reviewer_jR7t · 2025-08-06
> >
> > Thank you for your detailed rebuttal and the additional experiments you conducted during the response period. I really appreciate the effort—below I would like to offer a few clarifications and suggestions in response to your points:
> >
> > Q1: I strongly recommend polishing Section 2.2 to make it more detailed and accessible. Clearer exposition would help readers better understand your methodology.
> >
> > Q2: I appreciate the new analysis. While it may appear sufficient, I believe additional empirical evidence would strengthen your argument significantly. Currently, your discussion still feels overly analytical. In particular, using the observation that uniform performs well in your setup does not fully justify the general claim made in Section 2.4. That result might be confounded by your particular hyperparameters or experimental setting, and thus does not conclusively support such a **broad conclusion**—especially given that this discussion is central to the contribution of your paper.
> >
> > Q3: The additional experiments are helpful and much appreciated. I encourage you to include these results in the appendix and briefly reference them in the main text to improve completeness and transparency.
> >
> > W3: While you added new experiments to address the concerns with Table 4, it seems that the original results were somewhat glossed over in your rebuttal. Ideally, these new results should be presented as a supplement that strengthens the original findings—not as a replacement.
> >
> > Q5: I find your response here less convincing. It seems you're attributing the weak performance of DoReMi and DoGE to their fundamental limitations. While I understand that these methods may have certain drawbacks, these explanations sound a bit post-hoc. My concern is not about whether TANDEM can outperform them—I believe it can—but rather, why uniform performs better than these two **widely used** baselines. Are these methods really that fragile? Given how surprising this result is, I hope you understand my concern. This is not meant as nitpicking or skepticism, but rather reflects a genuine confusion given the reputation of these baselines. I remain worried about whether the implementations were faithfully executed.
> >
> > Overall, I think the core idea of your paper is both elegant and interesting, and the experimental results—especially after the rebuttal—are fairly comprehensive. **That said, I still hope the presentation can be improved, and I believe some of the current claims rely too heavily on speculative reasoning without sufficient empirical grounding, which slightly impacts the overall soundness.**
> >
> > Lastly, I apologize for the delayed response—I had some unexpected obligations. Thank you again for your thoughtful work. Given the improvements and clarifications in the rebuttal, I’m inclined to raise my score to a 3, though I may still take some time to consider and possibly discuss with other reviewers before finalizing.

---

> ### Author Response · Authors · 2025-08-07
> **Discussion Follow-up (1).**
>
> Thanks for recognizing our work as **elegant, interesting, and comprehensive in experiments**. We appreciate your insightful comments and would like to take this valuable opportunity to provide a more detailed discussion.
>
> > Q1:  Polishing Section 2.2 to make it more detailed and accessible.
>
> Thank you for your suggestion. We will thoroughly polish this section. Here we briefly explain the equation (3) $\sim$ (5). This section is about solving the penalized single-level objective by alternatively optimizing  $\boldsymbol{u}$, $\boldsymbol{w}$, and  $\boldsymbol{\alpha}$. Recall the penalized single-level objective:
>
> $\min _ {\boldsymbol{\alpha}\in\mathcal{A}, \boldsymbol{w}} \mathcal{H} _ {\gamma}(\boldsymbol{\alpha}, \boldsymbol{w}):= \mathcal{L} _ {\text{val}}( \boldsymbol{w})+ \gamma \left(\mathcal{L} _ {\text{train}}( \boldsymbol{\alpha}, \boldsymbol{w}) - \min _ {\boldsymbol{u}} \mathcal{L} _ {\text{train}}(\boldsymbol{\alpha}, \boldsymbol{u})\right)$
>
> * Update of $\boldsymbol{u}$ in Equation~(3) directly comes from $\min_{\boldsymbol{u}} \mathcal{L}_{\text{train}}(\boldsymbol{\alpha}, \boldsymbol{u})$.
>
> * Terms depend on $\boldsymbol{w}$ is $\mathcal{L} _ {\text{val}}( \boldsymbol{w})+ \gamma \mathcal{L} _ {\text{train}}( \boldsymbol{\alpha}, \boldsymbol{w})$. Applying SGD on which results in the update rule in Equation~(4).
>
>
> * For $\boldsymbol{\alpha}$, in the objective (2), only the $\gamma \left(\mathcal{L} _ {\text{train}}( \boldsymbol{\alpha}, \boldsymbol{w})- \min _ {\boldsymbol{u}} \mathcal{L} _ {\text{train}}(\boldsymbol{\alpha}, \boldsymbol{u})\right)$ is related to $\boldsymbol{\alpha}$. To see how this objective induces the update rule Equation (5), we first inspect the training loss. By definition, it is the $\boldsymbol{\alpha}$ weighted domain-wise loss: $\mathcal{L} _ {\text{train}}(\boldsymbol{\alpha}, \boldsymbol{w}) = \sum _ {m=1}^M \boldsymbol{\alpha} _ m\mathcal{L} _ {\text{train}}^{m}(\boldsymbol{w})$. The gradient w.r.t $\boldsymbol{\alpha}$ is exactly the domain-wise loss: $\nabla _ {\boldsymbol{\alpha}}\mathcal{L} _ {\text{train}}(\boldsymbol{\alpha}, \boldsymbol{w}) = \mathcal{L} _ {\text{train}}^{1:M}(\boldsymbol{\alpha}, \boldsymbol{w})$.  After the $K$ steps update of $\boldsymbol{u}$ and $\boldsymbol{w}$, substitute $\boldsymbol{u} _ k$ and $\boldsymbol{w} _ k$ results in the update rule Equation (5). $\Pi _ {\mathcal{A}}(\cdot)$ projects the updated $\boldsymbol{\alpha}$ into a probabilistic simplex. Running equation (3), (4) and (5) gives the solution of the bilevel DMO solution as shown in Theorem 1.
>
> > Q2: Some of the current claims rely too heavily on speculative reasoning. Additional empirical evidence would strengthen your argument significantly.
>
> Our "broad conclusion" is that Uniform effectively constitutes a valid solution for the bilevel optimization in data-abundant one-epoch training scenario. We are delighted to hear that **our current analytical description is sufficient**. Let us provide more empirical evidence now. Proposition 1 shows that as long as the per-domain generalization gap $\left|\mathcal{L} _ {\text {val}}^m-\mathcal{L} _ {\text {train }}^m\right| \rightarrow 0$, the Uniform weighting emerges as a valid solution (proof given in Appendix B). The following thing is to show that $\left|\mathcal{L} _ {\text {val}}^m-\mathcal{L} _ {\text {train}}^m\right| \rightarrow 0$ holds in the data-abundant one-epoch training, while not in the data-restricted/multi-epoch training.
>
> We conduct experiments in all the three scenarios and show the change of $\left|\mathcal{L} _ {\text {val}}^m-\mathcal{L} _ {\text {train }}^m\right|$ along training (As figure is not allowed during the dicussion, we temporally outline the results in tables, figures will be given in the revision.)
>
> For Data-Abundant Scenario:
> |Step Num.| step 0 | step 500 | step 1000 | step 1500 | step 2000| step 2500 |step 3000| step 3500 | step 4000 | step 4500 |
> |----------|--------|--------|--------|--------|--------|--------|--------|--------|--------|--------|
> |Arxiv| 0.0157 | 0.0203 | 0.0303 | 0.0152 | 0.0351 | 0.0221 | 0.0174 | 0.0116 | 0.0078 | 0.0344 |
> |Book| 0.2790 | 0.1268 | 0.0867 | 0.0331 | 0.0498 | 0.0095 | 0.0047 | 0.0144 | 0.0133 | 0.0227 |
> |C4| 0.0224 | 0.0100 | 0.0250 | 0.0179 | 0.0130 | 0.0102 | 0.0268 | 0.0158 | 0.0155 | 0.0007 |
> |CommonCrawl| 0.0346 | 0.0097 | 0.0128 | 0.0160 | 0.0213 | 0.0207 | 0.0167 | 0.0172 | 0.0066 | 0.0052 |
> |Github| 0.0963 | 0.0946 | 0.0816 | 0.0717 | 0.1099 | 0.0583 | 0.0857 | 0.0487 | 0.0425 | 0.0388 |
> |Stackexchange| 0.0273 | 0.0212 | 0.0230 | 0.0456 | 0.0406 | 0.0559 | 0.0485 | 0.0136 | 0.0490 | 0.0237 |
> |Wikipedia| 0.0427 | 0.0686 | 0.0637 | 0.0657 | 0.0444 | 0.0709 | 0.0710 | 0.0747 | 0.0663 | 0.0386 |
> |Averaged Gen. Gap| 0.0740 | 0.0502 | 0.0462 | 0.0379 | 0.0449 | 0.0354 | 0.0387 | 0.0280 | 0.0287 | 0.0234 |

---

> ### Author Response · Authors · 2025-08-07
> **Discussion Follow-up (2).**
>
> For Data-Restricted Scenario:
> |Step Num.| step 0 | step 500 | step 1000 | step 1500 | step 2000| step 2500 |step 3000| step 3500 | step 4000 | step 4500 |
> |---------|--------|--------|--------|--------|--------|--------|--------|--------|--------|--------|
> |Arxiv| 0.2695 | 0.1996 | 0.3559 | 0.3350 | 0.4940 | 0.4844 | 0.5292 | 0.7163 | 0.7162 | 0.8836 |
> |Book| 0.0582 | 0.2329 | 0.2805 | 0.4862 | 0.4749 | 0.6201 | 0.8135 | 0.7902 | 0.9747 | 0.9728 |
> |C4| 0.0437 | 0.0432 | 0.0143 | 0.0466 | 0.0120 | 0.0212 | 0.0270 | 0.0255 | 0.0463 | 0.0314 |
> |CommonCrawl| 0.0020 | 0.0214 | 0.0017 | 0.0229 | 0.0072 | 0.0096 | 0.0015 | 0.0134 | 0.0008 | 0.0083 |
> |Github| 0.1769 | 0.1069 | 0.2480 | 0.2043 | 0.2261 | 0.4081 | 0.4320 | 0.4281 | 0.6378 | 0.6469 |
> |Stackexchange| 0.1028 | 0.1781 | 0.0938 | 0.2713 | 0.2526 | 0.3993 | 0.6098 | 0.5957 | 0.8193 | 0.7599 |
> |Wikipedia | 0.1003 | 0.1488 | 0.1580 | 0.2798 | 0.3405 | 0.4903 | 0.5472 | 0.7824 | 0.7270 | 0.9156 |
> |Averaged Gen. Gap| 0.1076 | 0.1330 | 0.1646 | 0.2352 | 0.2582 | 0.3476 | 0.4229 | 0.4788 | 0.5603 | 0.6026 |
>
> For SFT:
> |Step Num.| step 0 | step 500 | step 1000 | step 1500 | step 2000| step 2500 |step 3000| step 3500 | step 4000 | step 4500 |
> |--------------|--------|--------|--------|--------|--------|--------|--------|--------|--------|--------|
> |Text Ent.| 0.0012 | 0.0095 | 0.0126 | 0.0068 | 0.0133 | 0.0051 | 0.0160 | 0.0367 | 0.0366 | 0.0449 |
> |Ans Ver.| 0.0073 | 0.0396 | 0.0647 | 0.0963 | 0.1028 | 0.1448 | 0.1553 | 0.2027 | 0.2556 | 0.2684 |
> |Text Mat.| 0.0163 | 0.0097 | 0.0066 | 0.0174 | 0.0325 | 0.0429 | 0.0353 | 0.0343 | 0.0690 | 0.0843 |
> |Inf Ext.| 0.0182 | 0.0191 | 0.0396 | 0.1176 | 0.1076 | 0.1865 | 0.2222 | 0.2372 | 0.2899 | 0.2917 |
> |Word Sem.| 0.0350 | 0.0916 | 0.1016 | 0.2147 | 0.2918 | 0.3591 | 0.3897 | 0.4102 | 0.4342 | 0.4200 |
> |Text Cat.| 0.0208 | 0.0222 | 0.0210 | 0.0178 | 0.0100 | 0.0004 | 0.0065 | 0.0246 | 0.0277 | 0.0173 |
> |Averaged Gen. Gap| 0.0164 | 0.0319 | 0.0410 | 0.0784 | 0.0930 | 0.1232 | 0.1375 | 0.1576 | 0.1855 | 0.1878 |
>
> We see that in the data-abundant scenario, the generalization gaps of each domain do not change much and are kept close to 0, $\left|\mathcal{L} _ {\text {val}}^m-\mathcal{L} _ {\text {train }}^m\right| \rightarrow 0$ holds so that Uniform is a valid solution. In the data-restricted scenario, we see an increase in the generalization gap, particularly on small domains (Arxiv, Book, Wikipedia), because the sample repetition kicks in. In the large domains (C4, CommonCrawl) where there is no data repetition, the generalization remains small.
> SFT is of the same case. Generalization gap $\left|\mathcal{L} _ {\text {val}}^m-\mathcal{L} _ {\text {train }}^m\right|$ increase in small tasks (Word Sem., Inf Ext.), while kept small for large tasks (Text Cat., Text Mat, Text Ent.). The results on
> **All three scenarios validate our analysis.**
>
> We believe this empirical result, plus the analysis in our rebuttal, well supports our argument and addresses your concern. Our claim is consistent with the results found in [1] that no existing DMO methods consistently outperform Uniform (in data-abundant scenario) in terms of average test perplexity per group (In abstract).
>
> >W3: While you added new experiments to address the concerns with Table 4...
>
> There might be a misunderstanding. We add the new SFT experiments (larger and more diverse) in response to question 3 for reviewer Vq6v, who is concerned that the original SFT tasks are small and lack diversity. We add it here for better comprehensiveness.  We also explain why the result in Table 4 is not that significant. The tasks are rather small (6000 samples per task), final metric for the 6 tasks is inconsistent with the loss. While TANDEM significantly decreases the test loss, the performance doesn't improve proportionally.

---

> ### Author Response · Authors · 2025-08-07
> **Discussion Follow-up (3).**
>
> > Q5: Why are DoReMi and DoGE not as good as Uniform?
>
> We fully understand your concern. After careful inspection, we confirmed that the performance of DoReMi is restricted by the a priori trained reference model. Following previous works, in all our experiments, we train the reference model on training data obtained using the Uniform strategy (as mentioned in the paper). This reference model, however, is not well suited for the data-restricted scenario, as the many small domains overfit and cannot provide faithful signals for DMO. Not that this is the default setting of all previous works.
>
> For a more comprehensive comparison, we train another reference model with the natural data mixture ratio (See Figure 4, the data statistics). This setting ensures no severe overfitting happens. The result is as follows:
> | Method           | 160M  | 410M  | 1B    |
> |------------------|-------|-------|-------|
> | Uniform          | 31.53 | 29.59 | 29.91 |
> | Natural          | 30.97 | 27.82 | 27.30 |
> | DoReMi-Natural   | 30.83 | 27.26 | 26.07 |
> | TANDEM           | 28.07 | 25.00 | 24.35 |
>
> We will make this clear and report this result to avoid misunderstanding in the revision.
>
> **Actually, the instability of DoReMi and DoGE has also been observed in many previous works[1,2,3]**.
>
> *  [1] presents that DoReMi underperforms Uniform (Table 1, the "Average" line, DoReMi-10k) .
> *  [2] shows that the perplexity of DoReMi can be significantly higher than Uniform in some cases (Table 2, column 2 (GH/C4), column 5 (CC/GH/W). In Table 19, GitHub data constitutes 46.7% of the training set with DoReMi, which is highly skewed. Usually, upweighting Common Crawl and C4 instead of Github leads to lower perplexity.
> *  [3] figures out that DoReMi is unable to produce competitive sampling weights for any domain, as it often gives the majority of the weight to a single source(Section 5.2, the Findings box).
> * From Table 2 in Aioli [2], DoGE can also underperform Uniform in many cases.
>
> So, it is normal that DoRemi and DoGE underperform Uniform in some cases. We believe the implementations were faithfully executed.
>
> Here again, we outline the weakness of these two baselines:
>
> * **DoReMi:** The problem of DoReMi lies in that it depends on a pre-trained static reference model. Imagine that in the data-restricted scenario and SFT, the reference model overfits on certain small domains (performs extremely low training loss, so the excessive losses are high), DoReMi may incorrectly continue to up-weight the domains to reduce the corresponding excessive losses. Besides, we know that, different from pretraining where the training loss decreases steadily, in SFT, there are always sudden downward jumps in training loss when the data repeats [4]. There are also chances that the SFT loss goes even to 0 for a large model. These pose greater challenges for DoReMi to obtain a good reference model.
>
> * **DoGE:** For SFT, DoGE suffers because it relies on the inner product of per-domain gradients, which exhibits large variance (SFT) and is difficult to provide reliable $\boldsymbol{\alpha}$ update directions (Figure 8). On the contrary, our TANDEM does not rely on the gradient estimation and utilizes $K$ steps probing to get more reliable $\boldsymbol{\alpha}$ updates.
>
>
> Once again, thank you for your valuable feedback and constructive suggestions. Sincerely hope that our clarifications help alleviate your concerns and contribute to a more favorable reassessment of our work.
>
>
> ------------
>
> [1] Simin Fan and Matteo Pagliardini, and Martin Jaggi. DOGE: Domain Reweighting with Generalization Estimation. ICML 2024.
>
> [2] Mayee F. Chen, Michael Y. Hu, Nicholas Lourie, Kyunghyun Cho, and Christopher Ré. Aioli: A unified optimization framework for language model data mixing. ICLR, 2025.
>
> [3] J. Parmar et.al. Data, Data Everywhere: A Guide for Pretraining Dataset Construction. EMNLP 2024 oral.
>
> [4] Jeremy Howard and Jonathan Whitaker. Can LLMs learn from a single example? 2023

---

### Official Review · Reviewer_3KfP · 2025-07-10

**Clarity:** 3
**Significance:** 3
**Originality:** 3
**Rating:** 5
**Confidence:** 3

**Summary:**

This paper tackles the problem of reweighting domains for language model pretraining. The key insight is to cast the problem as a bilevel optimization problem and used penalty-based to solve using gradient updates (as opposed to requiring Hessian information). Theoretically, they prove convergence to a first-order stationary point. Empirically, the method is shown to outperform previous methods such as DoReMi and DoGE for data limited settings (for normal settings, the uniform baseline is hard to beat) and SFT.

**Questions:**

Given that the u and w are meant to be close to each other, what happens if you run the same algorithm but with one copy (u = w)?

line 127: typo "constrain problem" => "constrained problem"

line 157: "by solving bi-level problem" => "by solving the bi-level problem", etc.

**Ethical Concerns:**

["NO or VERY MINOR ethics concerns only"]

**Limitations:**

yes

**Quality:**

3

**Strengths And Weaknesses:**

The problem of automatically selecting data mixes is of critical importance in language model pretraining. The paper presents a principled formulation based on bilevel optimization and show a practical algorithm to actually solve the problem. The formulation nicely and automatically captures important intuitions about what happens in the data limited regime and as the model size grows.

It is unclear whether the experimental settings reflect real-world LM training regimes. For example, 300M is a very small number of tokens for a 1B model (Chinchilla optimal would be already 20B tokens). Of course we are sometimes data limited but only on some domains, so is it possible to set up the experiment to have a much more sizable token count but have some rare domains that really matter for downstream evals? Also, given that most small models are overtrained (past Chinchilla), I'd be curious how the different methods perform; does careful mixing still matter? I am also quite curious about how this scales to 7B parameters and beyond, but of course I don't expect the authors to necessarily have the compute.

---

> ### Author Rebuttal · Authors · 2025-07-31
>
> We sincerely appreciate your recognition of our technical contributions. We address your concerns below:
>
> >Q1. Does careful mixing still matter when there is more sizable token count?
>
> Thanks for this insightful question. Generally, DMO is important as long as we are seeking a powerful and balanced model, even there are huge amount of tokens. For the ease of understanding, assume that different model capabilities emerge from different domains and are governed by their scaling laws each (neglecting the inter-domain relationships). DMO is effective as long as the benefits of scaling up certain domains outweigh the losses incurred by scaling down the other domains. Besides improving the overall model performance, it can be especially critical to balance the different capabilities.
>
> To validate this claim, we train the 160M models on the 6B version of sampled SlimPajama, which constitutes an overtrained (past Chinchilla) case:
> |Method|Arxiv|Book|C4|CC|Github|Stackexchange|Wikipedia| Average|
> |:---:|:---:|:---:|:---:|:---:|:---:|:---:|:---:|:---:|
> |Uniform|10.28| 32.37|38.53|34.57|5.86|10.76|15.26|17.09|
> |TANDEM|10.26|29.59|32.52|29.52|6.01|10.68|16.02|16.25|
>
> We see that TANDEM still outperforms Uniform. Though it seems that the improvement ($\sim$ 1)  is not as significant as that in the 300M data case ($\sim$ 3.5), we want to clarify that as the training proceeds and the perplexity goes down, further gains become inherently harder to achieve. This result is still significant, showing that careful mixing still matters in this overtrained case.
>
> > Q2. Experiments on larger models.
>
> Currently, the model scales (160M/410M/1B for data restricted case and 0.5B for SFT) are chosen following previous works, referring DoReMi (280M/510M/760M/1B), DoGE (60M/82M/124M), Skill-it (125M/1.3B) and Aioli (160M). The most recent work, released in May 2025, CHAMELEON, as mentioned by Reviever SkSi, also uses an 86M model for DMO. Nevertheless, we agree that scaling up the experiments would significantly strengthen the persuasiveness of our findings.
>
> Due to the limited computation and time, we conduct DMO with 3B models in the data-restricted scenario (Pythia 2.9B) and SFT (Qwen-2.5-3B). Scaling up to 7B+ models necessitates engineering upgrades to fit the two models ($\boldsymbol{u}$ and $\boldsymbol{w}$) within the 80GB memory, while also demanding substantial time and computational resources, so we resort to 3B experiments. We skip the data-abundant scenario as discussed in Section 2.4, Uniform is already a valid solution. The results are as follows:
>
> **Data-restricted scenario:**
> |Method|Arxiv|Book|C4|CC|Github|Stackexchange|Wikipedia| Average|
> |:---:|:---:|:---:|:---:|:---:|:---:|:---:|:---:|:---:|
> |Uniform|17.20|53.83|60.14|55.45|9.85|22.08|41.76|31.03|
> |TANDEM|15.73|42.85|43.60|39.59|7.49|17.26|29.20|23.81|
>
> We see that as the model goes larger, inappropriate mixture ratios (Uniform) lead to more obvious negative outcomes(e.g., much more severe overfitting on the overly sampled small domains). While TANDEM consistently generates proper mixture ratios.
>
> **SFT:**
>
> As suggested by the reviewer Vq6v, we use a larger SFT dataset. (Please refer to our response to question 3 for reviewer Vq6v)
> | Method  | Textual Ent. | Answer Ver. | Text Mat. | Inf. Ext. | Word Sem. | Text Cat. | Avg. Metric $\uparrow$ | Test Loss $\downarrow$ |
> |:-------:|:-----------:|:----------:|:--------:|:--------:|:--------:|:--------:|:-------------:|:-----------:|
> | Uniform | 91.9        | 74.8       | 88.8     | 79.2     | 88.8     | 85.5     | 84.9          | 0.189       |
> | TANDEM  | 92.4        | 76.9       | 89.6     | 80.1     | 88.0     | 87.9     | 85.8          | 0.174       |
>
> We see that for a larger 3B model, TANDEM still outperforms the Uniform baseline, showing its effectiveness.
>
> >Q3: What happens if you run the same algorithm but with one copy (u = w)?
>
> We can not simply set $\boldsymbol{u} = \boldsymbol{w}$. Actually, TANDEM deliberately deals with these two models. They are supposed to be close to each other as imposed by the Lagrange penalty, while they can not be exactly the same, as it is exactly the difference between them that motivates the update of $\boldsymbol{\alpha}$ (Equation (5)). So, if we maintain just one copy $\boldsymbol{u} = \boldsymbol{w}$, TANDEM will not return the optimized mixture ratio as expected.
>
>
> >Q3: Typos.
>
> Thanks very much for your careful checking, we've fixed these typos and thoroughly polished the revision.
>
> We thank the reviewer again for the review. If you have any more questions about our replies, or any further questions or concerns, please let us know.

---

> > ### Comment · Reviewer_3KfP · 2025-08-08
> >
> > Thanks for the response and the additional experiments, which address some of my questions.  I maintain my high score.

---

> > > ### Author Response · Authors · 2025-08-08
> > >
> > > Dear Reviewer,
> > >
> > > We sincerely thank you for your continued support and positive endorsement of our work throughout the review process. Your early recognition of our work as principled and practical has been a strong source of encouragement for us.
> > >
> > > Best regards,
> > >
> > > The Authors

---

> ### Author Response · Authors · 2025-08-04
> **Thank you again for your review.**
>
> Dear reviewer:
>
> Thank you again for your time reviewing our paper. We would appreciate it if you could confirm that our responses address your concerns. We would also be happy to engage in further discussions to address any other questions that you might have.
>
> Best regards

---

### Note · Authors · 2025-08-11

Dear Reviewers and Area Chairs,

We sincerely thank you for the thoughtful feedback. We are encouraged that all reviewers recognized the merits of our work:

* An **elegant, interesting, and theoretically grounded framework**.

* **insightful analysis** that unifies/improves previous works and discuss **when and why** DMO matters..

* **Comprehensive experiments** across various scenarios.

However, some of reviewer jR7t’s comments still appear to be based on a misunderstanding, despite our point-by-point responses.

1. Our claim, "Uniform is a solution for the bi-level DMO in the data-abundant scenario"  is not speculative but follows a clear chain of reasoning:

* **Uniform**   $\rightarrow$   **IID train/val data:** Follows the definition in Eq. 1.

* **Abundant (no repeated) IID data** $\rightarrow$ $\mathcal{L} _ {train} \approx \mathcal{L} _ {val}$: Holds for the first epoch training (analogies comparing the losses of two batches). **Supported by empirical evidence** in discussion.

* $\mathcal{L} _ {train} \approx \mathcal{L} _ {val}  \rightarrow $ **Uniform is a solution**:  Proposition 1, Appendix B

This is also supported by [2], showing no DMO methods consistently outperform Uniform in data-abundant scenario (In the abstract).

2. Regarding DoReMi and DoGE underperform Uniform in some cases. Our rebuttal detailed their weaknesses and explained why each scenario poses particular challenges. **The instability of DoReMi and DoGE has also been observed in previous works[1,2,3].**

* [1] presents that DoReMi underperforms Uniform (Table 1, DoReMi-10k) .
* [2] shows that PPL. of DoReMi can be significantly higher than Uniform. (Table 2, column 2 & 5. In Table 19, DoReMi is unreasonably skewed towards GitHub)
* [3] figures out that DoReMi is unable to produce competitive sampling weights for any domain (Section 5.2).
* From Table 2 in [2], DoGE also underperforms Uniform in many cases.

So, it is normal that in our experiments, the two baselines underperform in some cases. We believe the implementations were faithfully executed.

We would be deeply grateful if the overall assessment is grounded in the paper's broader contributions.

Yours sincerely,

Authors

[1] S. Fan et.al. DOGE: Domain Reweighting with Generalization Estimation. ICML'24.

[2] M. F. Chen et.al. Aioli: A unified optimization framework for language model data mixing. ICLR'25.

[3] J. Parmar et.al. Data, Data Everywhere: A Guide for Pretraining Dataset Construction. EMNLP'24.

---

### Decision · Program_Chairs · 2025-09-17

**Decision:**

Accept (poster)

**Comment:**

The paper proposes TANDEM, a bi-level data mixture optimization framework solved via twin networks. It reformulates domain reweighting into a single-level penalized problem, where a proxy model is trained on primary data and a dynamically updated reference model is trained with additional data. The difference between the two serves as a signal to adapt mixture ratios, yielding a principled, theoretically grounded, and practically efficient algorithm. TANDEM provides convergence guarantees and shows broad applicability across three scenarios: data-abundant pretraining, data-restricted pretraining, and supervised fine-tuning (SFT). Experiments demonstrate consistent improvements over strong baselines, especially in restricted and SFT regimes.

### Strengths of the paper:

- The reviewers (3KfP, Vq6v, SkSi) highlight the elegance and theoretical grounding of reformulating bi-level optimization with twin networks, noting that it unifies and clarifies prior methods (DoReMi, DoGE) while providing provable convergence.

- The method is versatile and broadly applicable, validated across data-abundant, data-restricted, and SFT settings, making it more comprehensive than most prior work.

- The paper is technically solid, offering both rigorous proofs and extensive empirical evaluation with multiple model scales and domains.

- Reviewers appreciated the insightful analysis of when and why DMO matters, going beyond algorithmic design to field-level insights.

- The rebuttal added larger-scale (3B) experiments, more diverse SFT evaluation (99 tasks), downstream task results, sensitivity analyses, and cost comparisons, strengthening the submission.

### Weaknesses of the paper:

- Reviewer jR7t noted presentation issues in the methodology, particularly the derivation of Eq. (5) and speculative claims in Section 2.4. While improved in rebuttal, clarity could still be strengthened in revision.

- The evaluation primarily uses relatively small models (<1B, some 3B). While this is standard in DMO literature due to compute constraints, it raises questions about scalability to 10B+ industry-scale models.

- Some empirical results, especially in SFT (Table 4), showed subtle gains without statistical significance, and the surprising underperformance of DoReMi/DoGE compared to Uniform prompted concerns about baseline tuning. The rebuttal provided clarifications and additional experiments, but skepticism remains.

- The computational cost, while analyzed, still involves training two networks, which may be heavy for larger-scale application.

- Writing quality was flagged as uneven in places, with grammar/notation issues that should be carefully polished.

### Primary reasons for Accept (Poster)

The primary reasons for recommending Accept (Poster) are that TANDEM provides a principled, elegant, and theoretically grounded framework for data mixture optimization, offering both unification of prior approaches and practical improvements in underexplored settings (restricted data and SFT). Despite limitations in model scale and presentation, the rebuttal convincingly addressed reviewer concerns with new empirical evidence, downstream evaluations, and detailed clarifications. The contribution is timely, impactful, and technically rigorous, making it a valuable addition to the NeurIPS program.

### Summary of the discussion and rebuttal

The authors engaged thoroughly with reviewers. For R-3KfP, they validated DMO importance in overtrained regimes and added 3B-scale experiments, showing TANDEM remains effective. For R-Vq6v, they expanded SFT evaluation to 99 tasks and added downstream results (ARC, BoolQ, HellaSwag, etc.), confirming generalization. For R-SkSi, they clarified derivations, reported FLOPs, and compared against Chameleon, showing TANDEM’s superiority in restricted regimes. For R-jR7t, they polished methodological explanations, provided empirical validation of generalization gap dynamics, and clarified why DoReMi/DoGE may underperform Uniform. While some concerns remain about scale and clarity, the rebuttal upgraded multiple scores and established consensus on the paper’s merit. Overall, TANDEM is recognized as a solid, rigorous, and timely contribution, meriting acceptance as a poster.